# The dynamic-thermal structures of the planetary boundary layer dominated by synoptic circulations and the regular effect on air pollution in Beijing

Yunyan Jiang[*][1,2], Jinyuan Xin[**][*][1,2,3], Ying Wang[4], Guiqian Tang[1], Yuxin Zhao[3,5], Danjie Jia[1,2], Dandan Zhao[1,2], Meng Wang[1], Lindong Dai[1], Lili Wang[1], Tianxue Wen[1], Fangkun Wu[1]

[1] State Key Laboratory of Atmospheric Boundary Layer Physics and Atmospheric Chemistry (LAPC), Institute of Atmospheric Physics, Chinese Academy of Sciences, Beijing 100029, China

[2] University of Chinese Academy of Sciences, Beijing 100049, China

[3] Collaborative Innovation Center on Forecast and Evaluation of Meteorological Disasters, Nanjing University of Information Science & Technology, Nanjing, 210044, China

[4] College of Atmospheric Sciences, Lanzhou University, Lanzhou 730000, China

[5] Institude of Atmospheric Composition, Chinese Academy of Meteorological Science, Beijing 100081, China

[*] These authors contributed equally to this work.

* Correspondence: Jinyuan Xin; email: xjy@mail.iap.ac.cn; phone: (+86)010-62059568; address: #40 Huayanli, Chaoyang District, Beijing 100029, China

**Abstract.** To investigate the impacts of multiscale circulations on the planetary boundary layer (PBL), we have carried out the PBL dynamic-thermal structure field experiment with a Doppler Wind Profile Lidar, a microwave radiometer and a ceilometer from January 2018 to December 2019 in Beijing. We found that the direct regulatory effect of synoptic circulation worked through transporting and accumulating pollutants in front of mountains in the daytime. While the indirect effect of multiscale circulations worked through coupling mechanisms in the nighttime. The horizontal coupling of different direction winds produced a severe pollution convergent zone. The vertical coupling of upper environmental winds and lower regional breezes regulated the mixing and diffusion of pollutants by generating dynamic wind shear and advective temperature inversion. We also found that the dominated synoptic circulations leaded to great differences in PBL dynamic-thermal structure and pollution. The cyclonic circulation resulted in a typical multilayer PBL characterized by high vertical shear (600 m), temperature inversion (900 m) and an inhomogeneous stratification. Meanwhile, strong regional breezes pushed the pollution convergent zone to the south of Beijing. The southwesterly circulation resulted in a mono-layer PBL characterized by low vertical shear (400 m) and inversion (200 m). The westerly circulation leaded to a hybrid-structure PBL, and the advective inversion generated by the vertical shear of zonal winds. Strong environmental winds of southwesterly and westerly circulations pushed the severe pollution zone to the front of mountains. There was no distinct PBL structure under the anticyclone circulation. The study systematically revealed the appreciable effects of synoptic and regional circulations on PBL structure and air quality, which enriched the prediction theory of atmospheric pollution in the complex terrain.

**Keywords** Synoptic Circulation Types, Planetary Boundary Layer, Multiscale Circulations Coupling, Regional Breezes, Air Pollution

## 1. Introduction

Beijing megacity is the political, economic and cultural center of China. With the recent economic development and acceleration of urbanization, an increasing number of air pollution episodes have emerged and pose a direct threat to human health (Quan et al., 2014; Fu et al., 2014; Cheng et al., 2016; Song et al., 2017).

Large-scale atmospheric circulations play a leading role in the transportation, accumulation and dispersion of pollution and thus result in the day-to-day variation of air quality (Tai et al., 2010; Zhang, 2017; Wang et al., 2018). Zheng et al. (2015) explored the relationships between AOD and synoptic circulations and found that a uniform surface pressure field in eastern China or a steady straight westerly in the middle troposphere is typically responsible for heavy pollution events. Miao et al. (2017) specially targeted summertime synoptic types, indicating that the horizontal transport of pollutants induced by the synoptic forcing is the most important factor affecting the air quality of Beijing in summer. They also found that synoptic patterns with high-pressure systems located to the east or southeast of Beijing are the most favorable types for heavy aerosol pollution events. Leung et al. (2018) indicated that daily PM2.5 had strong positive correlation with temperature and relative humidity but negative correlation with sea-level pressure in northern China. The PM2.5-to-climate sensitivities results were applied to predict future PM2.5 due to climate change, and found a decrease of 0.5μg/m³ in annual mean PM2.5 in the Beijing-Tianjin-Hebei region due to more frequent cold frontal ventilation. Liu et al. (2019) found that the episodes of PM2.5 pollution over the Beijing-Tianjin-Hebei region in winter were related to weather patterns such as the rear of a high-pressure system approaching the sea, a high-pressure field, a saddle pressure field, and the leading edge of a cold front. Li et al. (2020) quantitatively analyzed the contributions of different large-scale circulations toPM2.5. Many approaches have been used to classify the synoptic circulations, which can be mainly divided into subjective and objective methods. Objective weather typing methods have the advantages of convenient operation, high objectivity and efficiency, hence they have been employed widely in recent years (Zhang et al., 2016; Ye et al., 2016; Miao et al., 2017). In this study, we adopt an objective Lamb-Jenkinson classification scheme to categorize the large-scale atmospheric circulations centered on Beijing. The Lamb-Jenkinson approach, which is confirmed that the categorization results have clear physical understanding, has been applied widely in previous studies (Huang et al., 2016; Liao et al., 2017; Yu et al., 2017).

In addition, Beijing is located in the North China Plain (NCP) region and surrounded by Yan and Taihang Mountains to the north and west, respectively (Fig. 1b). The Bohai Sea lies to the southeast and is approximately 150 km from Beijing. This semibasin topography blocks and decelerates the relatively weak southerly airflows (Li et al., 2007). Aerosol pollutants from southern provinces through regional transportation stagnate and converge in front of the mountains, leading to the accumulation zone of pollution. This unique geographic location and topography results in diurnal variations in the mountain-plain breeze (MPB) and sea-land breeze (SLB) under relative weak synoptic circulations. The SLB can penetrate deep into the mainland when it is blooming, and aerosol pollution transported previously over the sea could be recirculated to the Beijing-Tianjin-Hebei region (Liu et al., 2009; Miao et al., 2017; Bei et al., 2018). As Beijing is surrounded by mountains and relatively far from the Bohai Sea, the intensity of the MPB circulation is much stronger compared to the sea-land breeze circulation in Beijing (Chen et al., 2009; Miao et al., 2015a, b), especially when synoptic circulations dominate in Bohai areas. Miao et al. (2015b) found that the regional-scale MPB circulations can modulate aerosol pollution by lifting or suppressing PBL. Chen et al. (2009) found that the MPB played an important role in the vertical transportation and dispersion of pollutants via the mountain chimney effect.

The PBL structure is also a key factor affecting the distribution and intensity of pollutants in addition to the circulations. The thermal structure of the PBL determines the vertical dispersion of aerosols. In the daytime convective layer, air pollution tends to be mixed vertically and homogeneously because of intensified turbulence and eddies of different sizes by radiation (Stull, 1988). After sunset, the turbulence decays and a stable boundary layer forms with weak turbulence. A radiation inversion on the ground caps the pollutants and leads to the accumulation near the surface. Hu et al. (2014) found that westerly warm advection above the Loess Plateau was transported over the NCP and imposed a thermal inversion, which acted as a lid and capped the pollution in the boundary layer (Xu et al., 2019). The dynamic structure of the PBL, including wind shears and turbulence, can

modify air quality by influencing the dispersion and transport processes of air pollutants (Li et al., 2019). Zhang et
al. (2020) found that a much weaker vertical wind shear was observed in the lower part of the PBL under polluted
conditions, compared with that under clean conditions, which could be caused by the strong ground-level PM2.5
accumulation induced by weak vertical mixing in the PBL. In turn, the particulate matter can also affect the PBL
structure by scattering and absorbing of solar radiation, and lead to severe pollution by positive feedback (Petaja
et al, 2016; Li et al, 2017). Ding et al. (2016) suggested that black carbon enhanced haze pollution in megacities in
China by heating upper PBL and cooling surface. Lou et al. (2019) investigated the relationships between PBL
height and PM2.5 and indicated that the strongest anticorrelation occurred in the NCP region at 1400 Beijing
time.

To sum up, because of the unique topography and geographic location of Beijing, large-scale circulation

and regional-scale thermodynamic circulation both have appreciable impacts on PBL and air pollution. What are
the characteristics of PBL structure and the temporal and spatial distribution of pollution under different
circulation types, and how do the multiscale circulations jointly force the PBL structure to change when they
coexist are still unrevealed. Therefore, one objective of this study is to investigate the PBL dynamic-thermal
structure and the distribution of severe pollution area under the most frequent circulation types in Beijing. The
other primary objective is to further explore the synergetic effects of multiscale circulations on PBL and pollution
in detail. Since the weather typing approach is able to classify the synoptic circulations into different types and
the high vertical resolution remote sensing observations can measure the fine dynamic-thermal structures of PBL,
the objectives can be achieved by employing weather typing approach and remote sensing measurements as a
necessary first step. The remainder of this paper is organized as follows. Sect. 2 describes the instruments, data
and method. Sect. 3 classifies the synoptic circulation types and selects typical types as research objects.
Moreover, it further investigates how the coupling mechanism of synoptic circulations and regional-scale
circulations changes the dynamic and thermal PBL structure and air pollution. Sect. 4 discusses the improvements
on previous studies and summarizes the main findings.

## 2.   Data and Method

A PBL field observation experiment was performed from January 2018 to December 2019 basing on

multiple remote sensing devices, including Doppler Wind Profile Lidar, microwave radiometer (MWR) and
ceilometer in the courtyard of the Institute of Atmospheric Physics (39.6°N and 116.2°E), Chinese Academy of
Sciences, Beijing (Fig. 1b). We systematically probed the PBL dynamic structure, thermodynamic structure and the
vertical distribution of aerosols using the Lidar three-dimensional winds, the MWR temperature and humidity
profiles and the ceilometer backscattering coefficient respectively. The original remote sensing data, with high
temporal and spatial resolution, are fully capable to show the fine PBL dynamic-thermal structure. The reanalysis
data of mean sea level pressure (MSLP) and winds are used to depict the synoptic circulations, and winds from
hundreds of automatic weather stations to characterize the fine regional circulations. Thus, the synergistic
impacts of coexisting synoptic-scale and regional-scale circulations on the PBL dynamic-thermal structure and air
pollution in Beijing megacity can be well understood using the remote sensing and meteorological data in
combination with the Lamb-Jenkinson weather typing approach. The typical cases lasting two days in the same
weather type (C, SW, W and A) are on October 22 to 24, July 26 to 28, May 15 to 17 in 2019 and December 28 to
30 in 2018 respectively. Due to the algorithm limitations on the observation conditions, the data of backscattering
coefficient and temperature profiles are missing about 5 hours on July 27, 2019.

## 2.1 Remote sensing data

The ceilometer (CL31, Vaisala) BL-VIEW software derives the PBL height according to the minimum value of

the local backscatter gradient (Tang et al., 2015), basing on the assumption that the aerosol concentration in

mixing layer (ML) is close to constant and significantly larger than that in the air above (Steyn et al., 1999). The BL-VIEW algorithm excluded profiles with fog, precipitation or low clouds, therefore resulting in the missing value of attenuated backscatter coefficient on July 27, 2019 used in southwesterly circulation. The vertical resolution of the backscatter is 10 meters and the maximal detection range can reach 7.7 km. A full overlap is achieved by using the same telescope for transmitting and receiving so that the backscatter can be used from the first range gate (Münkel et al, 2007). This gives a clear advantage over other commonly used Automatic Lidars and Ceilometers that usually show great uncertainty in the range below 200–500 m (Kotthaus et al., 2018). Three possible PBL heights, with a temporal resolution of 10 minutes, can be output simultaneously to characterize the multiple aerosol layers structure according to the first three largest negative gradients of backscatter. The typical uncertainty of CL31 on attenuated backscatter coefficient is ±20 % and is ±200 m on PBL height determination compared with radiosonde and other active remote sensors (Tsaknakis et al., 2011).

A Windcube 100S scanning Doppler Lidar is used to measure the wind profiles basing on the Doppler shift of aerosol particulate backscatter signals. Dai et al. (2020) suggested that the Doppler Wind Profile Lidar is fully capable to measure three-dimensional winds by comparing with cup anemometer and sonic wind anemometer. The vertical measuring range is from 50 m to 3.3 km. Several scanning modes are available and the DBS (Doppler Beam Swinging technique) mode, which includes four LOS (lines of sight) spaced 90° apart with a fixed elevation angle and one vertical LOS, is selected to detect the profiles of winds. The vertical resolution of the profiles is 25 m and the temporal resolution is 20 s. The velocity uncertainty along each LOS is associated with carrier-to-noise ratio (CNR) for each measurement volume following the methodology from O'Connor et al. (2010). Typically, a threshold of −22 or −23 dB is used as a limit for the accepted uncertainty in the Lidar measurements (Gryning et al., 2016), which corresponds to an uncertainty of about 0.15 m s$^{-1}$ (Aitken et al., 2012; Suomi et al., 2017).

The temperature and relative humidity profiles in RPG-HATPRO MWR are determined by neural network (NN) algorithm, and the vertical resolution of the profiles is 10–30 m in the lowest 0.5 km, 40–90 m from 0.5 km to 2.5 km, 100–200 m from 2 km to 10 km, and the temporal resolution is 1 s. The MWR used in this study has been tested by comparing with radiosonde observations (Zhao et al., 2019). The systematic errors increase with altitude, and the MWR-retrieved temperature and relative humidity are of quite high reliability inside the PBL. The temperature biases and RMSEs are -2-0 °C and 1-2 °C under 2 km, and the minimum of biases and RMSEs are between 1 km and 2 km, less than 0.5 °C and 1.3 °C respectively. Since the relative humidity derived from the temperature and water vapor density, both the errors can cause the uncertainties. The bias and RMSE of relative humidity is about -5% and 15% under 2 km. Furthermore, the residual liquid droplets on the water film led to high brightness temperature measured by the MWR, resulting in the abnormal high values of the temperature and humidity data. Therefore, data on July 27, 2019 were eliminated and substituted with missing values.

## 2.2 Meteorological data

The daily MSLP and wind at 850 hPa from the National Center for Atmospheric Research (NCAR) reanalysis data (gridded at 2.5° × 2.5°) were used to classify the synoptic circulation patterns and depict the background circulations of the typical circulation types. The divergence and vertical velocity reanalysis data (gridded at 1° × 1°) with a temporal resolution of 1 h from Re-analysis Interim (ERA-Interim) of European Centre for Medium-Range Weather Forecasts (ECMWF) were used to study the vertical motion in the mid-low troposphere in the NCP region and its impact on PBL structure.. The hourly mean wind at the surface in the Beijing-Tianjin-Hebei area were collected by hundreds of automatic weather stations operated by the China Meteorological Administration (CMA).

## 2.3 Pollutant data

The hourly PM2.5 concentrations in the Beijing-Tianjin-Hebei monitoring sites are acquired from the National Urban Air Quality Real-time Publishing Platform (http://106.37.208.233:20035/) issued by the Ministry of

Ecology and Environment. There are 35 air quality monitoring stations in Beijing (Fig. 4a) and 68 monitoring sites in Tianjin and Hebei provinces (Fig. 5, 7, 9). The PM2.5 concentration in Beijing are shown in shaded by interpolating data of 35 sites, while the PM2.5 concentration in other areas are shown in scatter with color as the spatial resolution is relative low. The PM2.5 data of Olympic Center station, which is the closest monitoring site to the location of remote sensing measurements (less than 1 km), is used in the circulation classification.

## 2.4 Method

The Lamb-Jenkinson weather typing (LWT) approach is widely adopted in large-scale circulation classification (Lamb 1972; Jenkinson and Collison, 1977) because of its automation and explicit meteorologically meaning. To classify the synoptic circulation types, the daily MSLP in 2018 and 2019 were used. The LWT scheme is a half-objective categorization method. The weather patterns are predefined and each day can be identified objectively as one certain type according to a small number of empirical rules (Trigo and DaCamara, 2000). As shown in Fig. 1a, 16 gridded pressure data surrounding the study area (Beijing city) were selected to calculate the direction and vorticity of geostrophic wind. The synoptic circulation can be classified into 26 types in total including two vorticity types (cyclonic, C; anticyclonic, A), eight directional types (northeasterly, NE; easterly, E; southeasterly, SE; southerly, S; southwesterly, SW; westerly, W; northwesterly, NW; and northerly, N), and sixteen hybrid types (CN, CNE, CE, CSE, CS, CSW, CW, CNW, AN, ANE, AE, ASE, AS, ASW, AW, and ANW).

The gradient Richardson number (Ri) is the ratio of the buoyancy term to the shear term in the turbulent kinetic equation. A negative Ri is an indication of buoyancy-generated turbulence, while positive Ri less than 0.25 indicates shear turbulence and dynamic instability. When Ri is larger than 0.25 and less than 1.0 the flows become neutral, or exhibit hysteresis and still maintain turbulent. Otherwise, Ri larger than 1.0 means turbulent flow will turn to be dynamically stable laminar (Stull, 1988). The distributional characteristics of Ri can reveal whether the PBL has a stratified structure or not (Banakh et al., 2020). Thus, we adopt the critical values of 0.25 and 1.0 as a criterion to determine the PBL structure. Ri can be calculated by Equation 1, where g is the acceleration of gravity and $\Delta z$ is the height interval between adjacent layers. $\overline{\theta}$ is the mean virtual potential temperature, $\Delta\overline{u}$ and $\Delta\overline{v}$ is the mean zonal and meridional wind speeds within the height interval respectively.

$$R_i = \frac{\frac{g}{\overline{\overline{\theta}}}\frac{\Delta\overline{\theta}}{\Delta z}}{(\frac{\Delta\overline{u}}{\Delta z})^2 + (\frac{\Delta\overline{v}}{\Delta z})^2} \quad (1)$$

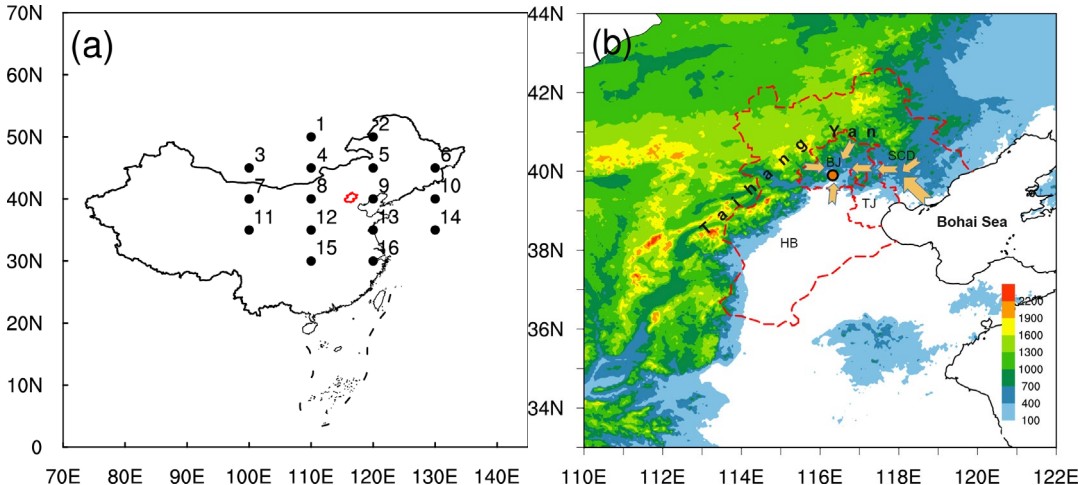

Fig. 1 The locations of 16 grid data of the 5°× 10° MSLP used for Lamb-Jenkinson weather type classification (black dots) (a). The terrain height of the North China Plain (shaded, units: m) (b). The location of Beijing city and the Beijing-Tianjin-Hebei region is marked by the red solid lines and red dashed lines respectively. The orange dot indicates the location of remote sensing devices. The arrows indicate the horizontal coupling mechanism of how multiscale circulations affect pollution by generating convergent zone.

## 3. Results and Discussions

### 3.1 The typical weather types and PM2.5 distribution

Based on the Lamb-Jenkinson weather typing approach, synoptic circulations from 2018 to 2019 were classified into predefined 26 circulation patterns and each day has a specific type. The distributional characteristics of daily averaged PM2.5 concentration, as well as the occurrence frequency of different circulation patterns, were statistically conducted. The occurrence frequencies of the two vorticity and eight directional types were much higher than those of the other sixteen hybrid types, accounting for 75% of total days (Fig. 2). According to the pollution intensity, three pollution types (cyclonic C, southwesterly SW and westerly W) and one clean type (anticyclonic A) occurring most frequently in the NCP were selected as the studied circulation patterns. It was consistent with the results of Li et al. (2020) on the relationship between pollutant concentration and circulation types in northern China. Weather types with high PM2.5 concentration but occurring no more than ten times, such as type CE and type CW, were not discussed in this article. The average and extreme PM2.5 concentrations of type C reached 77 $\mu g/m^3$ and 270 $\mu g/m^3$, respectively, and were much stronger than the other pollution types. Clearly, the cyclonic circulation pattern was more conducive to severe pollution events. The circulation of type A was the most common type, and the PM2.5 concentration was 28 $\mu g/m^3$, which was the lowest.

As shown in Fig. 3, the locations of the high and low pressures and the intensity of the wind fields at 925 hPa under different circulation patterns were clearly distinct. In type C, Beijing was located in the center of low pressure, and the sea to the east of China was controlled by an anticyclone (Fig. 3a). Southwesterly winds prevailed, flowing northward to Beijing along the periphery of the anticyclone with an average wind speed of 3 m/s. In type SW, Beijing lay southeast of the low pressure in Mongolia, and the high pressure over the sea was significantly enhanced compared with type C (Fig. 3b). Therefore, southeasterly winds prevailed to the south of Beijing and shifted southwesterly after flowing by. In type W, westerly winds were dominant and converged with southwesterly flows to the north of Beijing (Fig. 3c). The mean velocity of environmental flows in type SW and type W was observably larger than that in type C. In general, the mainland was mainly controlled by low pressure with an anticyclone lying over the sea to the east of China in pollution types C, SW and W, and southerly flows dominated at 925 hPa. By contrast, northern China in the clean type A was occupied by high pressure. Beijing was located in the center of high pressure with strong northerly winds in the lower level (Fig. 3d).

The pollution intensity is closely related to the large-scale weather circulations. Although the dominant synoptic patterns in different seasons vary greatly, the modulating effects on air pollution of specific circulation types in different seasons are similar (Liao et al., 2017; Li et al., 2020). The spatial distribution of PM2.5 in Beijing under pollution types C, SW, W and clean type A is shown in Fig. 4. Type C had the highest pollution level, with the PM2.5 concentration increasing from 60 $\mu g/m^3$ in the northwestern mountainous area to 90 $\mu g/m^3$ in the south-central plain area, which was significantly higher than the values for types SW and W. Type A was highly ventilated, with a PM2.5 concentration below 30 $\mu g/m^3$ in most areas. Under the influence of semibasin topography surrounded by mountains on three sides (Fig. 1b), the pollution concentrations in all weather types were characterized by a gradual decrease from southeast to northwest in Beijing.

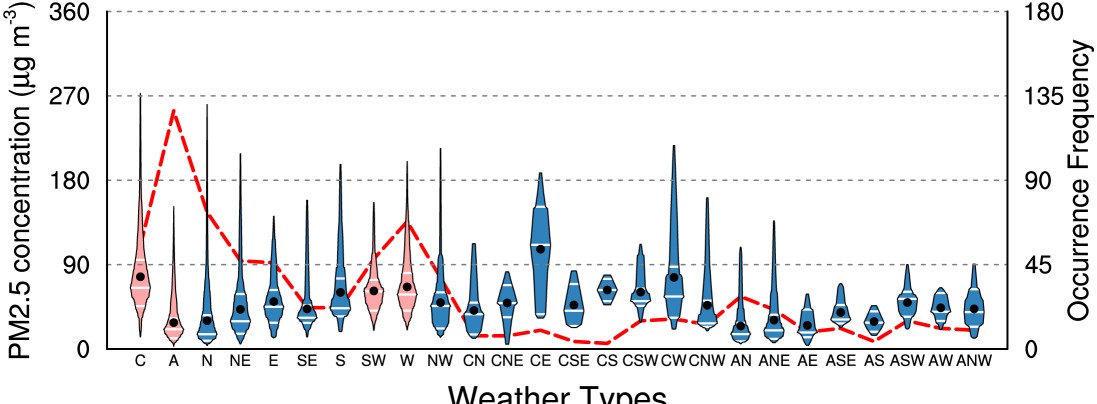


Fig. 2 Daily averaged PM2.5 concentration in Olympic Center station (box plots, units: $10^{-1}$ μg m$^{-3}$) and the
occurrence frequencies of 26 weather types (red dashed lines) from 2018 to 2019. The red boxes represent
classical types selected for research. The black dots represent the mean values.

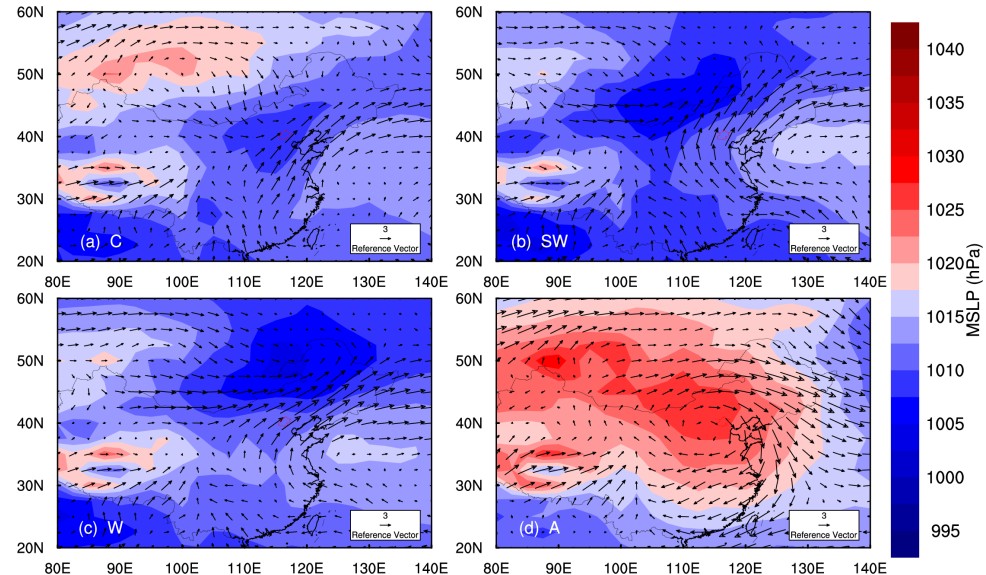


Fig. 3 The daily MSLP (shaded, units: hPa) and wind fields at 925 hPa (vectors, units: m s$^{-1}$) for types C (a), SW (b),
W (c) and A (d) from 2018 to 2019.

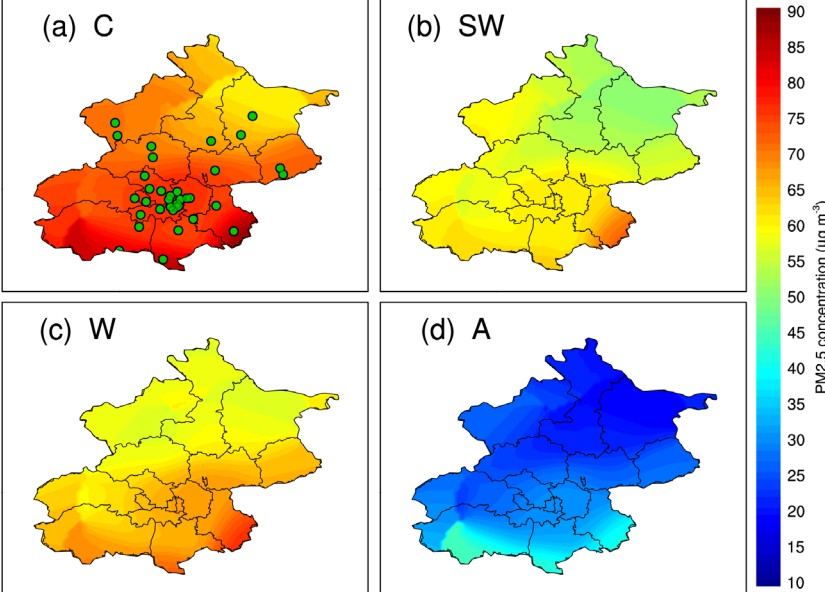


Fig. 4 The averaged PM2.5 concentration (shaded, units: $10^{-1}\,\mu g\ m^{-3}$) in Beijing for types C (a), SW (b), W (c) and A (d) from 2018 to 2019. The green dots in Fig. 4a indicate the locations of air quality monitoring sites in Beijing.

## 3.2 The flow field and dynamic-thermal structure of the PBL under typical weather types

As mentioned above, due to the special topography and geographical location in Beijing, both large-scale weather circulations and regional-scale thermal circulations have conspicuous effects on modulating pollution. In addition, the thermal and dynamic structure of the PBL also has an appreciable impact on the mixing and diffusion of pollutants. Therefore, the multiscale circulations can not only influence the pollution directly but also influence it by changing the PBL structure indirectly. To reveal the mechanisms of how the coupling effects of multiscale circulations affect the PBL structure and air pollution under different synoptic patterns, we conduct an analysis of the horizontal flow field and vertical PBL structure in depth by choosing typical cases lasting two days in the same weather type (C, SW, W and A). The typical cases are on October 22 to 24, July 26 to 28, May 15 to 17 in 2019 and December 28 to 30 in 2018 respectively.

### 3.2.1 Multilayer PBL structure under type C circulation

The mainland was governed by low pressure under type C synoptic circulations, and the ambient winds were mainly southwesterly (Fig. 3a). On the afternoon of 22nd, the plain breezes in central Hebei, which were induced by thermal contrast between the mountain and plain, blocked weak environmental winds and the direct transportation of pollutants to Beijing (Fig. 5a). The westerly and the northerly mountain breezes began to prevail at night while the conversion from sea breeze to land breeze was not obvious (Fig. 5b). The onshore winds in the coastal area were notably larger than the northerly mountain breezes in southern Chengde (SCD), which were diverted to the west and east. The diverted easterly winds converged with the onshore winds, enhancing the easterly winds and the east pollution transport channel. Sun et al. (2019) have found that the pressure gradients between the plain and mountain areas are critical causes of the easterly winds in Beijing. Consequently, easterly winds gathered with mountain breezes and formed a pollution convergent zone. Weak environmental winds not only made the pollution channels hard to establish but also caused the pollutants to recirculate southward by strong downslope breezes further in the early morning (Fig. 5c). A mesoscale convergent belt was generated in southeastern Hebei, providing conditions for the transportation of pollutants later. At noon on 23rd, the intensified plain winds transported high concentrations of aerosols from the right side of the convergent belt to Beijing (Fig. 5d). Large-scale environmental winds were strengthened and dominated in the afternoon (Fig. 5e), leading to the establishment of the south and east pollution transport channels and further exacerbating the air quality. On the night of 23rd, easterly winds were observably strengthened again, joining with the downslope breezes and the ambient southerly flows (Fig. 5f). The four directional airflows formed a convergent zone that caused pollutants to accumulate dramatically in the plain areas. This convergent region that is generated by the coupling effect of large-scale circulation and regional-scale mountain breezes at night also appeared in other pollution types, as will be discussed later.

The PBL under type C circulation presented a multilayer structure without diurnal variation (Fig. 6a). The highly stable structure and weak ambient winds resulted in a higher aerosol concentration near the surface than that in the other pollution types (Fig. 4). The pollution decreased from bottom to top within the PBL and was characterized by a gradient distribution. It is consistent with previous research (Jiang et al., 2021) that the top PBL height is equal to the maximum detection range of wind Lidar. In the daytime, environmental southwesterly winds dominated within the PBL. In the horizontal flow field, zonal winds from Tianjin to the southeast of Beijing turned to be easterly winds and the northerly downslope winds in Beijing were strengthened later on the night of 22nd (Fig. 5b, 5c). Inside the PBL, easterly and northerly winds extended to 600 m above the ground from 20 pm on 22nd to 10 am on 23rd (Fig. 6b, 6c), thus the directional shear of meridional and zonal winds increased

considerably. The shallower nocturnal PBL coincided with the zero speed zone between the upper environmental
winds and lower regional-scale breezes with the largest directional shear (Fig. 6b, c). Variations of the vertical
dynamic structure in the PBL drove the thermal structure to adjust. Warm air advected by large-scale
southwesterly winds overlay on the cold air advected by regional-scale northeasterly breezes. Consequently, a
conspicuous advective temperature inversion occurred near the shallower nocturnal PBL at 08-09 am on 22nd
and 00-11 am on 23, ranging from 600 m to 900 m above the ground (Fig. 6d). The Richardson number Ri away
from the temperature inversion structure was less than 0.25 (turbulent region) during the night, while it increased
considerably from the periphery of inversion and was larger than 1.0 (stable region) promptly. The sharp jump of
Ri from the turbulent region to the stable region of inversion indicated a vertical stratified structure inside the PBL.
The result suggested that the nocturnal PBL has an inhomogeneous stratification structure characterized by
strong variations of Ri accompanied by inversion structure (Fig. 6e). However, the relatively stronger northerly
breezes compared to the environmental winds made the pollutants recirculate southward horizontally, the wind
shear developed so high that the dynamically stable region was above 300 m and the inversion was above 600 m;
the pollutants dispersed vertically to some extent consequently (Fig. 5c, 6c). Compared to the previous night, the
ambient winds on the night of 23rd were stronger; thus, both south and east transport channels were established,
along with the pollution convergent zone (Fig. 5f). The weak easterly and northerly winds were lower than 300 m
(Fig. 6b, c), resulting in temperature inversion and stable stratification connected to the ground. A high
concentration of pollution was accumulated in the convergent zone horizontally and trapped below the lowest
PBL vertically. Thus, the PM2.5 concentration on the night of 23rd was significantly higher than that on 22nd.

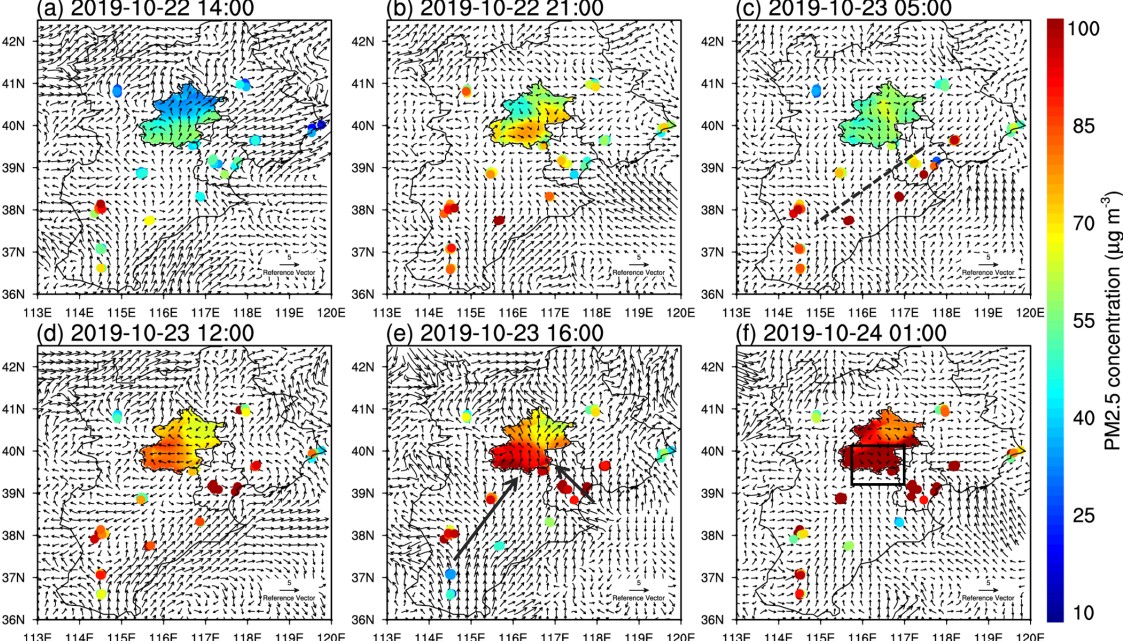


Fig. 5 The surface winds (vectors, units: m s$^{-1}$) in the NCP and PM2.5 concentration in Beijing (shaded, units: 10$^{-1}$
μg m$^{-3}$), Hebei and Tianjin monitoring sites (scatter, units: 10$^{-1}$ μg m$^{-3}$) of different times (Local Time) for type C.
The dashed line represents the convergence belt. The arrow lines represent the pollutant transport channels. The
rectangle represents the convergent zone.

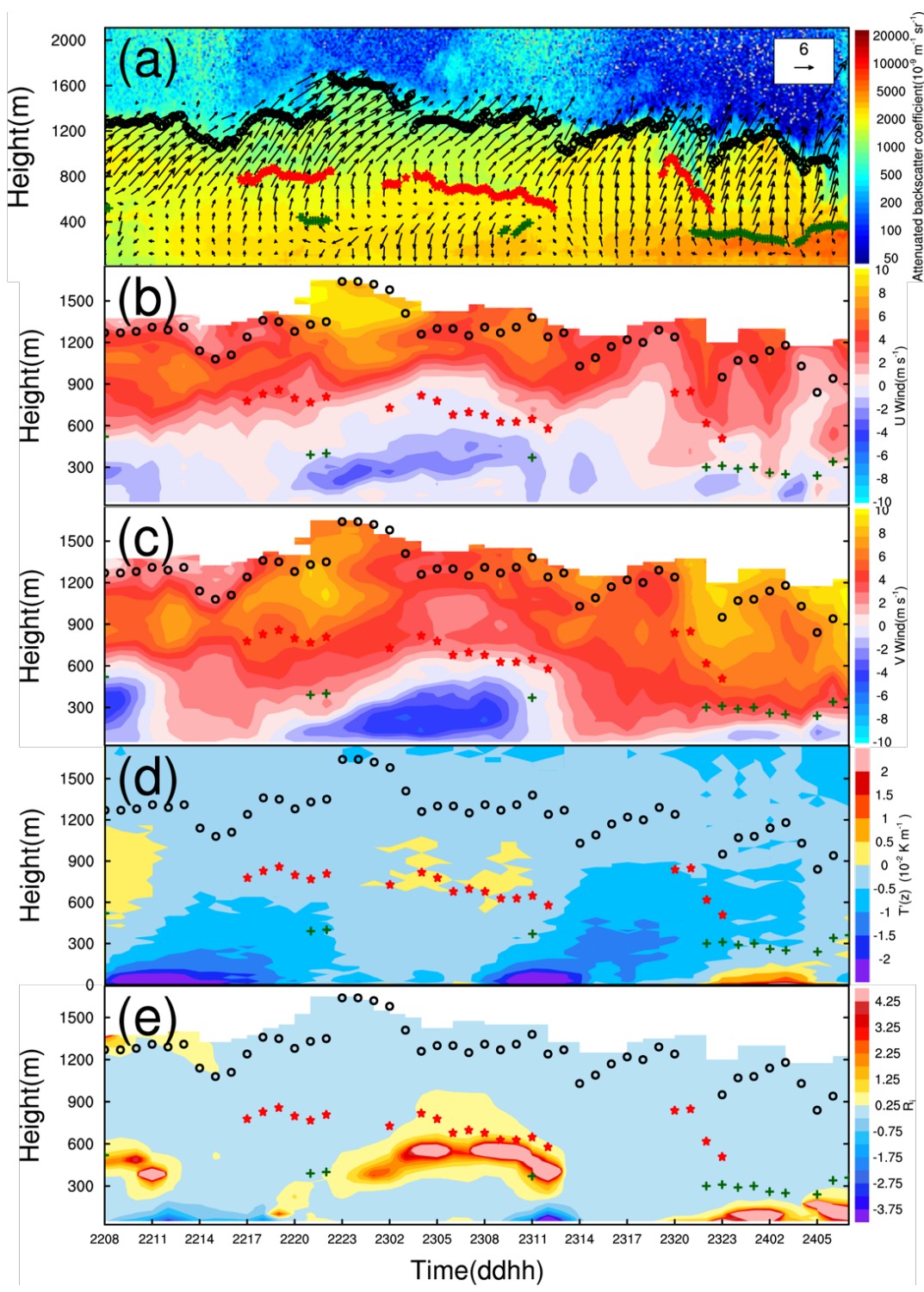

Fig. 6 Attenuated backscatter coefficient (shaded, units: $10^{-9}$ m$^{-1}$ sr$^{-1}$) measured by ceilometer and horizontal winds (vectors, units: m s$^{-1}$) measured by Lidar (a), zonal wind component speeds (shaded, units: m s$^{-1}$) (b), meridional wind component speeds (shaded, units: m s$^{-1}$) (c), gradient of temperature T'(z) (shaded, units: K km$^{-1}$) measured by MWR (d), and Richardson number (shaded) (e) for type C. The green crosses, red stars and black hollow dots represent the lowest, middle and top PBLH, respectively.

### 3.2.2 Mono-layer PBL structure under type SW circulation

Under type SW circulation, the easterly wind component increased in southeastern Hebei and the Bohai Sea, and the velocity of environmental winds was appreciably higher than that in type C. (Fig. 3b). On the early

morning of 26th, mountain breezes carrying clean air masses prevailed in Beijing, and the air quality was good (Fig. 7a). The basic southerly winds dominated in the Beijing-Tianjin-Hebei region in the afternoon, transporting pollutants northward and causing airflow to converge in plain areas (Fig. 7b). However, pollutants were ventilated horizontally by strong ambient winds and diffused vertically by the intensified turbulent mixing within the growing ML, so the aerosol concentration grew slowly during the day (Fig. 8a). At night, the mountain breezes were strengthened while the ambient southerly winds were weakened; hence, the pollutants were transported to Beijing via the east pollution channel (Fig. 7c). Multiscale circulations of different directions joined and generated a convergent zone in the plain area. Afterwards, easterly flows were further strengthened and transported pollutants to Beijing continuously, the severely polluted area moved westward (Fig. 7d, 8a). In the daytime of 27th, the ambient winds prevailed again, and strong ambient winds removed pollutants by enhancing the ventilation and turbulent mixing (Fig. 7e, 8a). Therefore, the PM2.5 concentration decreased instantly and the air quality in the Beijing-Tianjin-Hebei region improved markedly (Fig. 7f).

Unlike type C, the PBL presented a monolayer structure in type SW, and the aerosol within the PBL was uniformly distributed (Fig. 8a). Furthermore, the PBL had an obvious diurnal variation and the maximum detection distance of wind Lidar was only consistent with the top ML in type SW. The nocturnal PBL and the growing or collapsing ML were usually lower than the maximum detection distance, indicating that there were residual aerosols above the PBL. In the daytime of 26th, southwesterly winds dominated within the PBL, and the temperature lapse rate was greater than 0.5 °C/100 m. Along with radiation reinforcing turbulent kinetic energy, the PBL rose to 1200 m. Pollutants were transported to Beijing but mixed vertically (Fig. 8a), so the PM2.5 concentration near the surface grew slowly (Fig. 7b). On the night of 26th, the regional-scale circulation developed upward, and the vertical wind shears between the lower regional breezes and upper environmental winds were strengthened prominently (Fig. 8b, c). The warm advection overlay on the cold advection resulted in advective inversion, forcing the PBL to adjust to become stable (Fig. 8d). Correspondingly, Ri experienced an appreciable increase from the turbulent region above the PBL to the stable region of below the PBL (Fig. 8e). The nocturnal PBL has a homogeneous dynamically stable structure. Similar to type C, a high concentration of pollutants was trapped below the zero wind speed zone where the nocturnal PBL was located. In the daytime of 27th, large-scale environmental winds within the PBL were strengthened greatly. The PBL height was 800 m higher than that of the previous day; thus, the pollutants were advected horizontally and diffused vertically (Fig. 8a). The basic southerly winds with high speed prevailed in central and southern Beijing on the night of 27th, preventing the mountain winds from flowing southward (Fig. 7f). As a result, no vertical shear of meridional winds occurred in the dynamic field (Fig. 8c) and no temperature inversion occurred in the thermal field (Fig. 8d). The PM2.5 concentration was further reduced. It can be inferred that the temperature inversion in type SW was generated by the vertical thermal contrast of meridional winds. When the meridional winds were uniformly southerly winds within and above the PBL, the air masses in the upper layer had the same thermal properties as that in the lower layer, which will reduce the vertical wind shear and destroy the stable inversion structure.

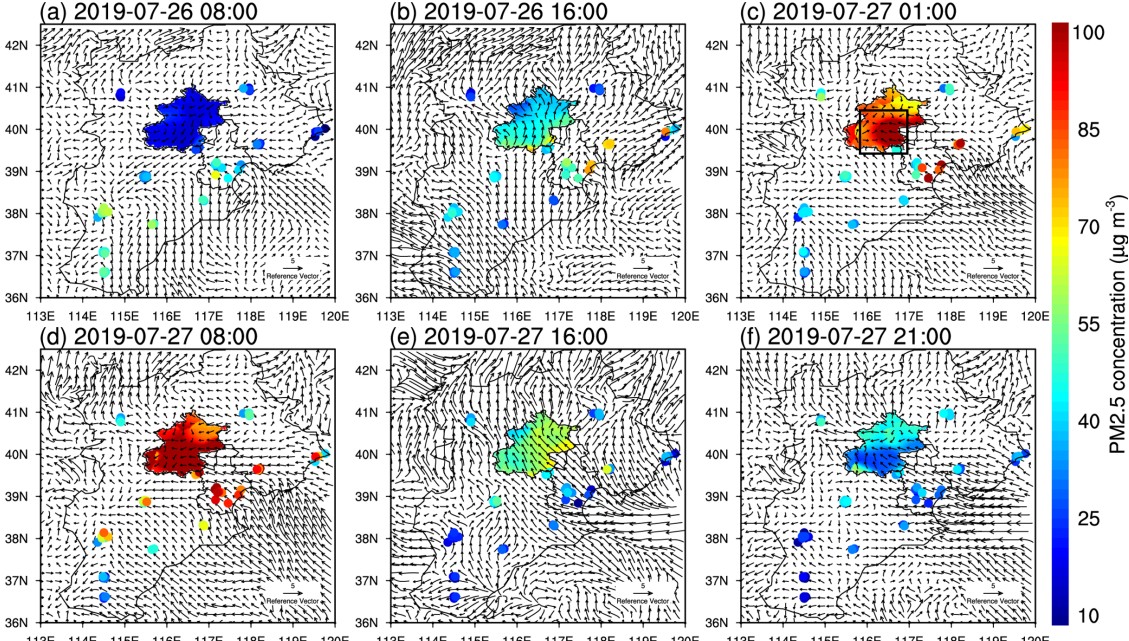

Fig. 7 The surface winds (vectors, units: m s$^{-1}$) in the NCP and PM2.5 concentration in Beijing (shaded, units: 10$^{-1}$ µg m$^{-3}$), Hebei and Tianjin monitoring sites (scatter, units: 10$^{-1}$ µg m$^{-3}$) of different times (Local Time) for type SW. The rectangle represents the convergent zone.

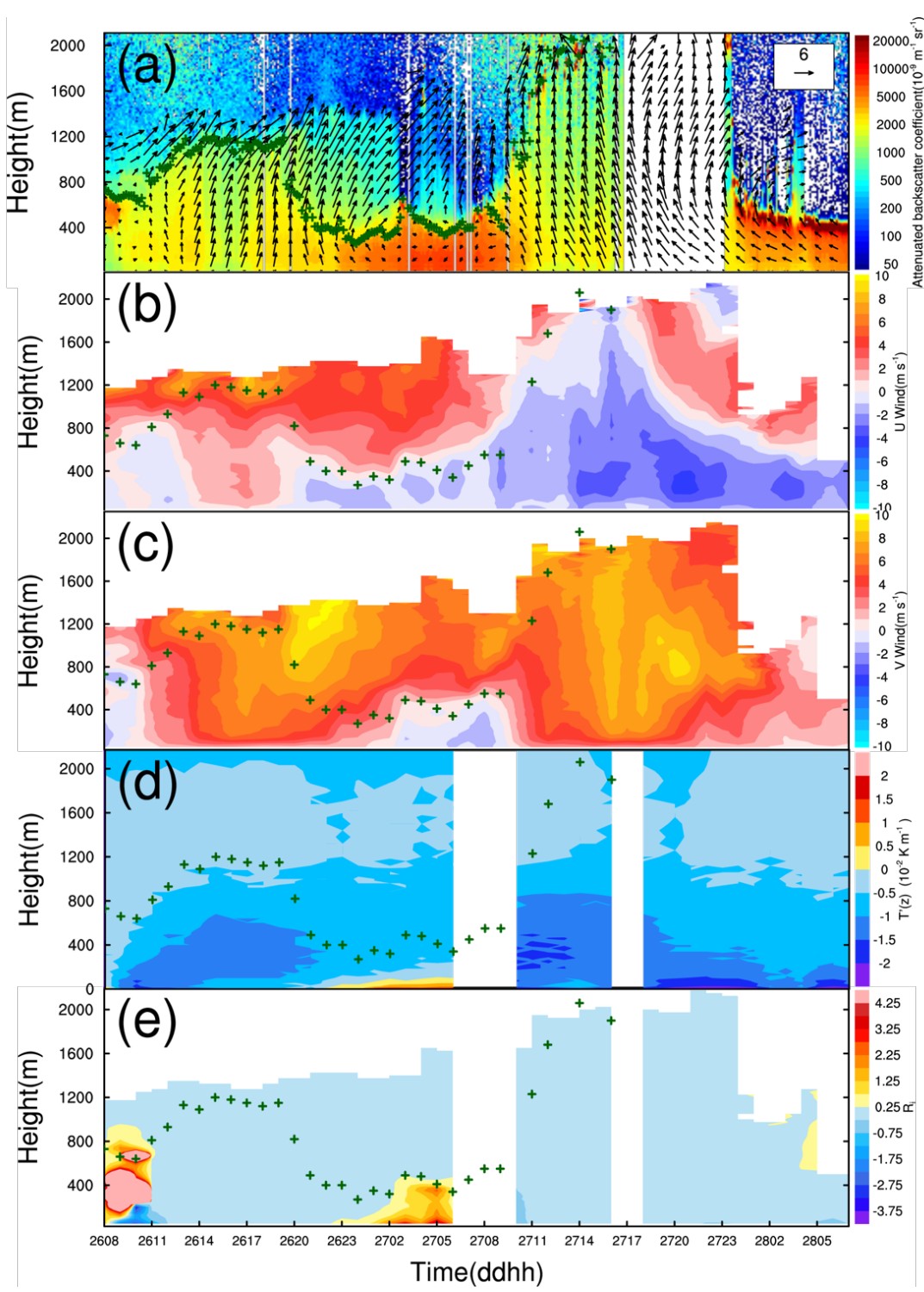

Fig. 8 Attenuated backscatter coefficient (shaded, units: $10^{-9}$ m$^{-1}$ sr$^{-1}$) measured by ceilometer and horizontal winds (vectors, units: m s$^{-1}$) measured by Lidar (a), zonal wind component speeds (shaded, units: m s$^{-1}$) (b), meridional wind component speeds (shaded, units: m s$^{-1}$) (c), gradient of temperature T'(z) (shaded, units: K km$^{-1}$) measured by MWR (d), and Richardson number (shaded) (e) for type SW. The green crosses represent the PBLH.

### 3.2.3 Hybrid structure PBL under type W circulation

Under type W circulation, strong easterly winds transported a high concentration of aerosols to Beijing through the east pollution channel, and the PM2.5 concentration had already reached a high level in the early morning (Fig. 9a). Taking the mountain as the boundary, environmental westerly winds prevailed in northwestern

Hebei and southwesterly winds prevailed in southern Hebei in the afternoon. The two directional flows carried pollutants and formed a convergent belt along the western mountains (Fig. 3c, 9b). This distribution of synoptic circulations in type W was conducive to the occurrence of severe pollution around mountains. Similar to other pollution types, the ambient winds converged with region-scale mountain breezes at night, forming a convergent zone (Fig. 9c). The convergent zone moved southward later because of intensified mountain breezes (Fig. 9d). The large velocity of environmental winds leads to strong ventilation (Fig. 9e). In addition, the increasing PBL made the pollutants diluted vertically, and the air pollution was alleviated temporarily. On night of the 16th (Fig. 9f), the synergistic effects of multiscale circulations led to the convergent zone again, and pollution occurred in the easterly flows with a high PM2.5 concentration.

The PBL under type W circulation presented a hybrid structure, having similar characteristics of types C and SW simultaneously. Similar to type C, the aerosol concentration was characterized by a gradient distribution within the multilayer PBL (Fig. 10a). However, the PBL had an obvious diurnal variation, and the maximum detection distance of wind Lidar was only consistent with the top ML in the daytime, similar to type SW. Although the PBL height reached 1600 m in the daytime (Fig. 10a), the PM2.5 concentration at the surface did not decrease observably because of the massive pollution accumulated previously and the continuous emissions and transportation of pollutants (Fig. 9b). The mixing layer collapsed along the zero wind speed of meridional winds after sunset, and the breezes within nocturnal PBL shifted northwesterly at night (Fig. 10b, c). In type W, zonal circulation dominated. The vertical shear of zonal winds was intensified significantly at night, while the vertical shear of meridional winds diminished. Therefore, it can be assumed that the temperature inversion in type W was produced by the vertical shear of zonal winds. The thermal contrast between the upper westerly winds and the lower easterly winds produced a deep inversion layer that existed from the surface to 500 m (Fig. 10d), as well as a dynamically stable structure with a depth exceeding 600 m (Fig. 10e). This is consistent with the findings of Hu et al. (2014) that westerly warm advection from the Loess Plateau was transported over the NCP and imposed a thermal inversion above the PBL. The top of the PBL was consistent with the top of the inversion and zero wind speed zone, and a high concentration of aerosols was trapped below the zero wind speed zone.

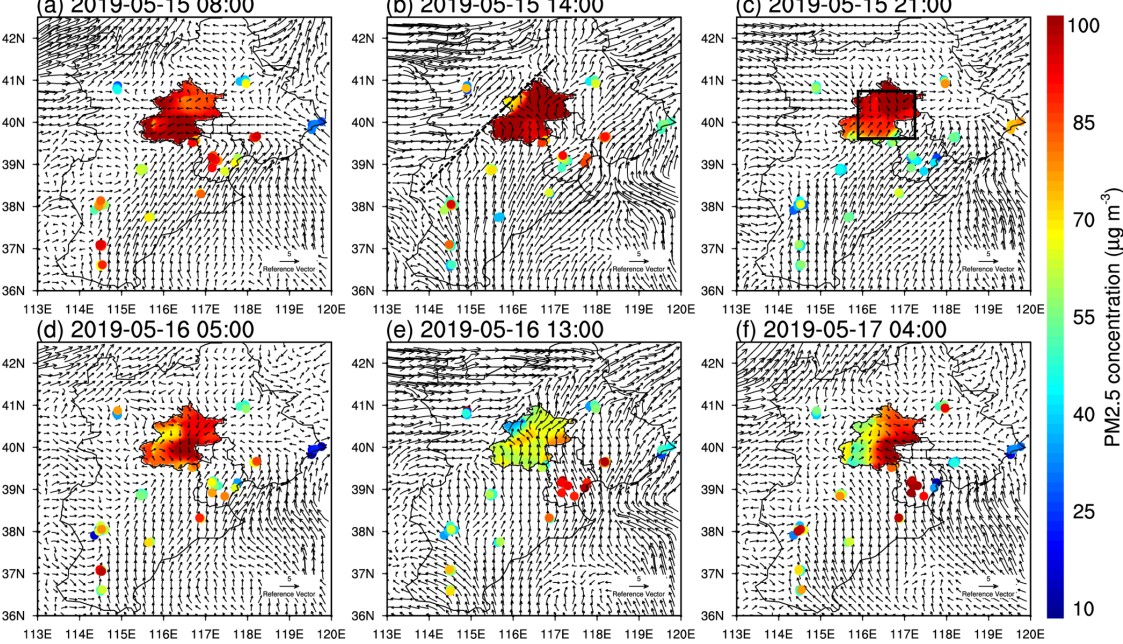

Fig. 9 The surface winds (vectors, units: m s⁻¹) in the NCP and PM2.5 concentration in Beijing (shaded, units: $10^{-1}$ μg m⁻³), Hebei and Tianjin monitoring sites (scatter, units: $10^{-1}$ μg m⁻³) of different times (Local Time) for type W. The dashed line represents the convergence belt. The rectangle represents the convergent zone.

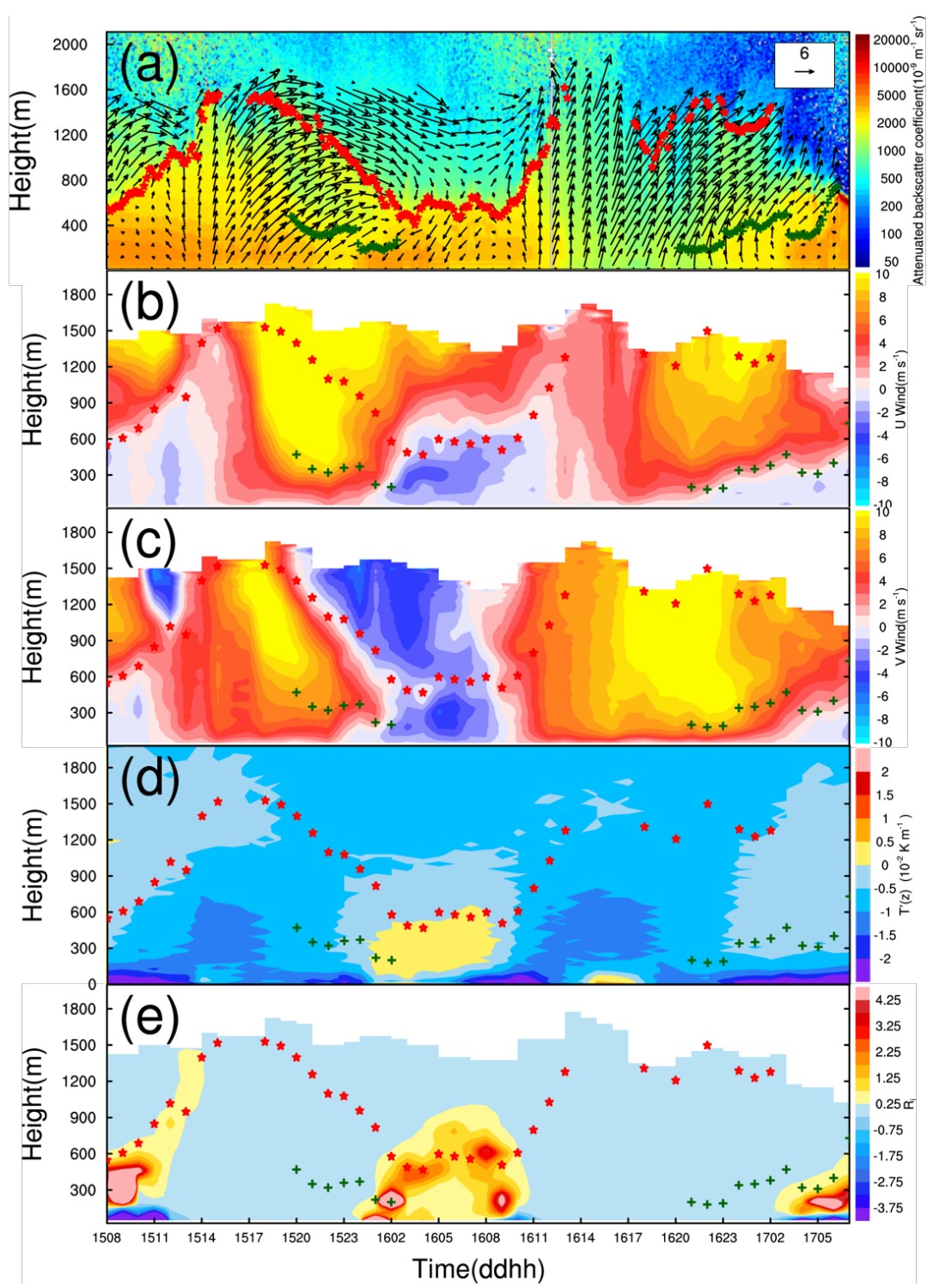

Fig. 10 Attenuated backscatter coefficient (shaded, units: $10^{-9}$ m$^{-1}$ sr$^{-1}$) measured by ceilometer and horizontal

winds (vectors, units: m s$^{-1}$) measured by Lidar (a), zonal wind component speeds (shaded, units: m s$^{-1}$) (b),

meridional wind component speeds (shaded, units: m s$^{-1}$) (c), gradient of temperature T'(z) (shaded, units: K km$^{-1}$)

measured by MWR (d), and Richardson number (shaded) (e) for type W. The green crosses and red stars represent

the low and top PBLH, respectively.

## 3.2.4 Strong turbulent PBL structure under clean type A circulation

Strikingly different from the circulations of pollution types, the mainland was under high pressure control in

the clean type, and northwesterly winds with a high velocity carrying clean air masses moved southward (Fig.

11a). Strong winds were favorable for the turbulent mixing and the vertical dispersion of pollutants. In addition, the strong ventilation was beneficial to the horizontal spreading of pollutants. Due to the intense turbulent mixing, the vertical wind shear and the diurnal variation of thermal field disappear, and there is no distinct PBL structure different from the free atmosphere (Fig. 11a, b). The lapse rate of temperature was greater than 1 °C/100 m, and Ri was less than 0.25 within the PBL (not shown). Although the aerosol concentration of the clean type was far less than that of pollution types, the PBL height was only 500 m at night (Fig. 11a). Sometimes, the PBL in the clean type was even lower than that of pollution types, or extended to 2-3 km swiftly because of the instant upward diffusion of aerosol particulates. Unlike pollution types, the PBL height is inconsistent with the maximum detection range of wind Lidar. Therefore, different circulation types should be distinguished when analyzing the long-term relationships between the PBL height and pollution concentration. As shown in Fig. 12 c and d, under the governing of high pressure, descending and divergent airflows of the clean type dominated the whole lower and middle parts of the troposphere, and the sinking velocity was significantly higher than that of pollution types. The vertical velocity changed little vertically due to the northerly winds with a large speed penetrating downward. The intensity of sinking and divergence was higher at night than that in the day, with the strongest divergence occurring near the surface.

## 3.3 Multiscale circulations coupling mechanism for air pollution

In addition to horizontal circulations, the vertical motion of basic airflows is also a crucial dynamic factor in forming stable structure during pollution episodes. The pollution types shared similar vertical motion characteristics as shown in Fig. 12. The basic flows at the bottom of the troposphere is convergence and the flows above it is divergence at all times of a day (Fig. 12b). In the daytime, the environmental southerly winds ware obstructed on three sides by mountains. Airflows slowed down or stagnated in the plain areas, forming the topographic convergence. While at night the convergence was caused by the joint of environmental winds and regional breezes, and the height of convergence zone reduced simultaneously with the nocturnal PBL because the regional circulations developed below the shallower nocturnal PBL. Unlike the divergence field, the vertical velocity in the daytime differed from that in the nighttime because of the diurnal variations of PBL structure (Fig. 12a). In the daytime, the thermodynamic convection and the wind speed were enhanced expressively (Fig. 8a, 10a), thus the intensified turbulence will help the flows to move upward and cause the pollutants close to the ground to mix vertically within the PBL to some extent. However, the sinking and divergent flows superposed above the PBL, preventing the pollutants from moving upward continuously and making it difficult for the aerosol particulates to diffuse beyond. As a consequence, the pollutants accumulate slowly in the daytime because of the common influences of horizontal topographic blocking and vertical upward mixing with the increasing PBL. However in nighttime, as the thermodynamic convection weakened and the inversion structure formed, it turned to be sinking movement at the bottom of the troposphere when the cold northerly regional breezes prevailed. Wu et al. (2017) found that the descending motion of synoptic circulations contributed to a reduction in the PBLH by compressing the air mass. Therefore, massive pollutants were capped near the surface and accumulated rapidly at night under the convergent sinking motion accompanied by temperature inversion structure.

To sum up, different pollution patterns (C, SW and W) have similar influential mechanisms that both horizontal and vertical coupling effects of the multiscale circulations have contributed to air pollution. The horizontal coupling mechanism is shown in Fig. 1b. The environmental winds transport pollutants emitted from southern sources to Beijing, mainly through south and east pollution channels. Large-scale environmental winds and regional-scale breezes are coupled, generating a convergent zone of four directional flows horizontally and aggravating the air pollution directly at night. The relative strength of winds makes the severely polluted area move around horizontally from 39°N to 41°N. The schematic of Fig. 13 demonstrates that the vertical coupling mechanism further influences the mixing and dispersion of pollution indirectly by changing the PBL structure. In

the daytime, the sinking divergent flows overlaying the rising convergent flows within the PBL inhibit the continuous upward dispersion of pollutants. At night, the warm advection transported by the upper environmental winds overlies the cold advection transported by the lower regional breezes, generating strong directional wind shear and advective inversion, which are near the top of regional breezes. This dynamic structure forces the PBL to be a stable stratification. The nocturnal PBL is located at the zero speed zone between the regional-scale breezes and the environmental winds, and the relative strength of winds determines the PBL height. The capping inversion cooperating with the convergent sinking motion within the PBL suppresses massive pollutants below the zero speed zone.

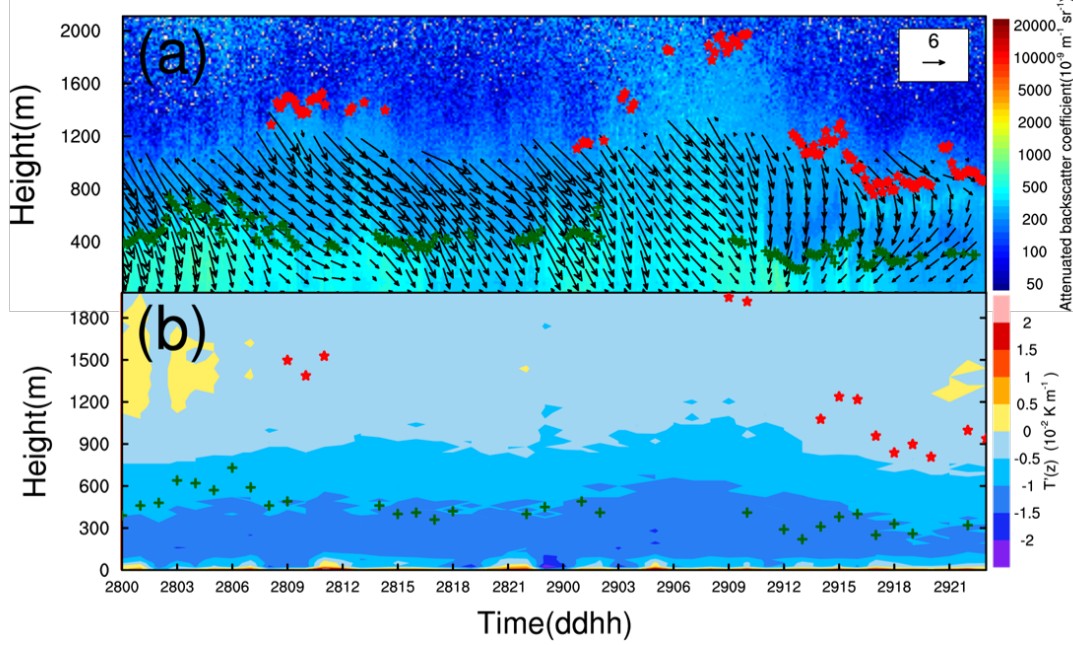

Fig. 11 Attenuated backscatter coefficient (shaded, units: $10^{-9}$ m$^{-1}$ sr$^{-1}$) measured by ceilometer and horizontal winds (vectors, units: m s$^{-1}$) measured by Lidar (a), and gradient of temperature T'(z) (shaded, units: K km$^{-1}$) measured by MWR (b) for type A. The green crosses and red stars represent the low and top PBLH, respectively.

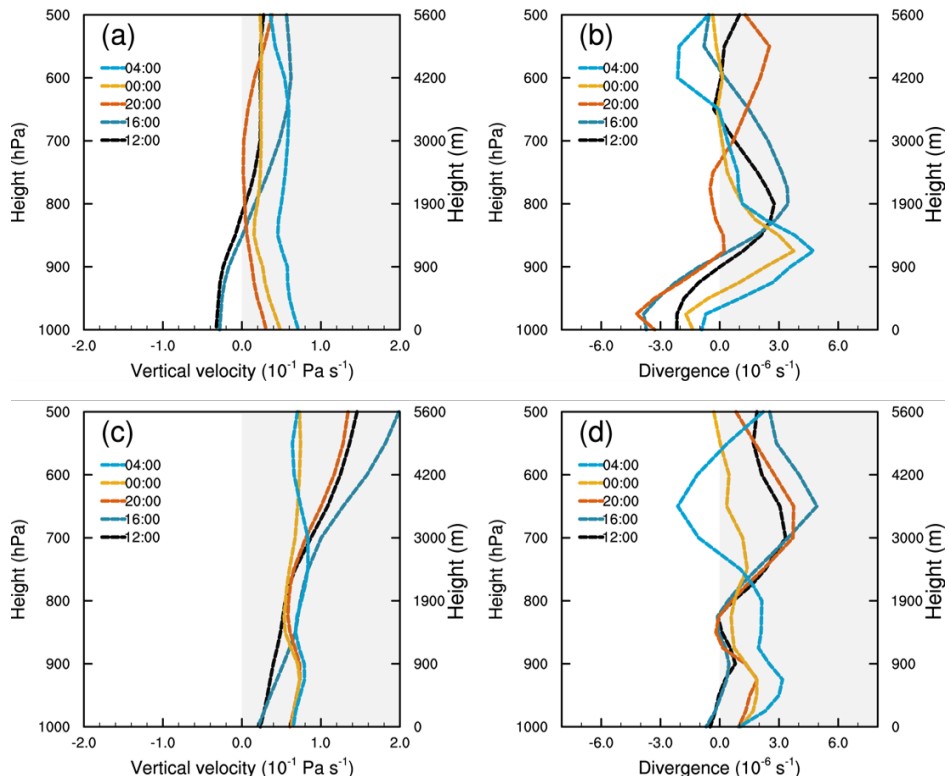


Fig. 12 The averaged vertical velocity (units: Pa s$^{-1}$, negative (positive) value denotes updraft (downward)
movement) (a, c) and divergence (units: $10^{-5}$ s$^{-1}$) (b, d) of pollution types (a, b) and the clean type (c, d) in the
North China Plain

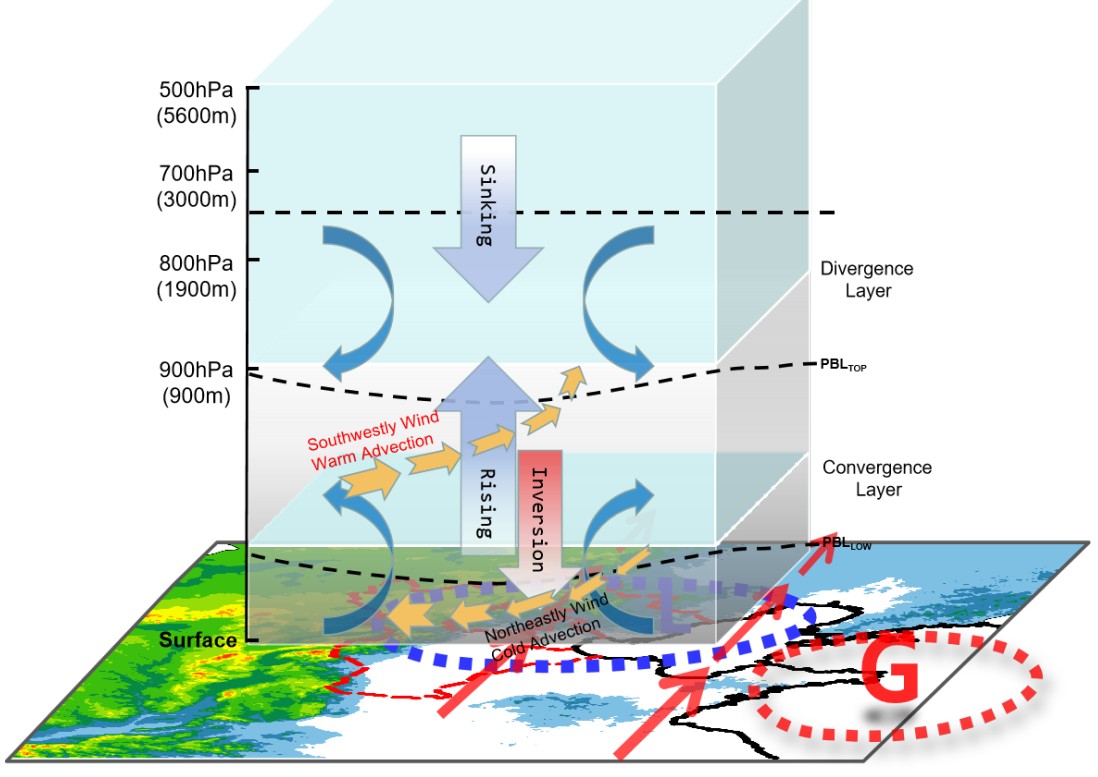


Fig. 13 The schematic of vertical coupling mechanism of multiscale circulations for typical pollution types; The
horizontal part is the background circulations of MSLP. The vertical part is the PBL dynamic-thermal structure over
the NCP region

## 4.  Conclusions and Discussion

This paper explores the direct regulatory effect and indirect coupling effect of synoptic circulations by choosing the most frequent pollution types and clean type classified by LWT approach. The PBL dynamic-thermal structure and the severe pollution area under typical circulations types are further investigated. Results suggest that different pollution patterns have similar influential mechanisms on PBL structure and air pollution. The direct regulatory effect of synoptic circulations plays a leading role in the daytime, large-scale southerly winds dominate and are favorable for the pollution transport to NCP region and the accumulation in front of mountains in the early stage of pollution. However during the period of pollution, the relative stronger southerly winds and the increasing PBL height are adverse to the accumulation of pollutants, or even make pollutants ventilated horizontally and diluted vertically. While the indirect effect played a leading role in the nighttime by coupling mechanisms. The coexisting multiscale circulations at night, on the one hand, affect the pollution via the horizontal coupling effect, which produces a pollution convergent zone of different direction winds. The relative strength of winds makes the polluted area move around horizontally between 39°N and 41°N. On the other hand, the multiscale circulations regulate the mixing and diffusion of pollutants by the vertical coupling effect, which changes the PBL dynamic and thermal structure. Vertical shear between the ambient winds and regional-scale breezes leads to advective inversion structure with strong variations of Ri. The nocturnal shallower PBL is consistent with the zero velocity zone, where massive pollutants were suppressed below, and the relative strength of winds determines the PBL height.

The multilayer PBL under type C circulation has no diurnal variation. Weak ambient winds strengthen the mountain breezes observably at night, as a result the vertical shear and temperature inversion can reach 600m and 900 m respectively. An inhomogeneous stratification with sharp jump of Ri is formed from the periphery of inversion. The severe polluted area was located to the south of Beijing. The mono-layer PBL under southwesterly circulation with obvious diurnal variation can reach 2000 m in the daytime. Strong environmental winds restrain the development of regional breezes at night, the zero speed zone is located at 400 m and the inversion generated by the vertical shear of meridional winds is lower than 200 m. Southerly winds within and above the PBL having the same thermal properties will diminish the vertical shear and damage the advective inversion structure. The PBL under westerly circulation has a hybrid structure with both multiple aerosol layers and diurnal variation. The inversion is generated by the vertical shear of zonal winds. The polluted areas under southwesterly and westerly circulations are located more northerly. Clean and strong north winds are dominated under anticyclone circulation, the vertical shear and the diurnal variation of thermal field disappear and there is no distinct PBL structure.

This study suggests that synoptic-scale circulations or the regional-scale circulations don't influence the PBL structure and air pollution separately but by the synergistic ways instead. The new knowledge of the coupling mechanism of multiscale circulations has appreciable implications for deepening the understanding of cooperation of influential factors in severe pollution processes in the background of unique topography. The new findings about the PBL dynamic-thermal structure and the distribution of pollution provide a reference for forecasting the severe pollution area under the most frequent synoptic circulation types in Beijing. Although the essential impacts of synoptic-scale and regional-scale circulations on PBL dynamic-thermal structure are emphasized in the paper, the feedback impact of aerosols should not be neglect either when investigating the PBL structure and air pollution.

## Data availability

The hourly ground level PM2.5 concentration data can be obtained from the National Urban Air Quality

Real-time Publishing Platform (http://106.37.208.233:20035/). Other data used in this study can be acquired upon
request to the corresponding author.

## Competing interests

The authors declare that they have no known competing financial interests or personal relationships that
could have appeared to influence the work reported in this paper.

## Author contribution

XJ designed the study. JY, WY, TG, JD, ZD, WM, DL WL, WT, WF contributed to observation data, provided
experimental assistance and analyzed methodology. JY and XJ wrote the paper with inputs from all the other
authors.

## Acknowledgments

This study was supported by the National Key Research and Development Program of China (grant number
2016YFC0202001) and the Chinese Academy of Sciences Strategic Priority Research Program (XDA23020301).

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
