# Peer review of "The dynamic-thermal structures of the planetary boundary layer dominated by"

_Atmospheric Chemistry and Physics, 2020_

## Referee Comment (RC1) · Anonymous Referee #1 · 5 Nov 2020

Based on Lamb-Jenkinson weather typing and multiple field measurements, this study reveals the mechanisms which couple multiscale circulations, planetary boundary layer (PBL) structure and air pollution. Due to the topographic blocking during daytime, pollutants accumulate in the plain areas within different layers. The sinking divergent flows overlying on the rising convergent flows within the PBL inhibit the continuously upward dispersion of air pollutants. At night, the horizontal and vertical coupling mechanisms increase the air pollution. The large-scale wind systems and regional-scale breezes affect the air pollution directly via the horizontal coupling, which generates air pollution

convergent zones of different directional flows. The strength of flows causes severely polluted areas from 39°N to 41°N. In addition, the multiscale circulations regulate the mixing and diffusion of pollutants indirectly via the vertical coupling, which changes the PBL dynamic-thermal structure. The warm advection caused by the upper winds overlies the cold advection caused by the lower regional breezes, generating strong wind direction shears and advective inversions. The blocking inversion and the convergent sinking motion within the PBL suppress massive air pollutants below the zero speed zone. The multilayer PBL under cyclonic circulation has no diurnal variation. Weak ambient winds strengthen the mountain breezes observably during night, the temperature inversion can reach 900 m. The nocturnal shallower PBL, consistent with the zero velocity zone between ambient and mountain winds, can reach 600 m. Otherwise, the PBL under southwesterly circulation is a mono-layer with obvious diurnal variation, reaching 2000 m during daytime. The strong circulations restrain the development of regional breezes, the zero speed zone is located at 400 m and the inversion is lower than 200 m during night. The PBL under westerly circulation has a hybrid structure with multiple air pollution layers and diurnal variation. The inversion is generated by the vertical shear of zonal winds. Clean and strong northerly winds dominate under anticyclone circulation, the vertical shear and the diurnal variation of thermal fields disappear because of strong turbulent mixing, and there is no significant PBL structure. Our results imply that the algorithms of description of atmospheric environmental capacity under synoptic circulations, such as the cyclonic type, with a multilayer PBL need to be improved. General comments The impression of the whole paper is a description of mechanisms of coupling atmospheric dynamics and air pollution levels in the North China Plain (NCP) and especially in the Beijing area. The introduction is missing the definition of objectives of the paper and maybe some hypothesis of answers for questions. The description of methods is missing an overall statement which data are required and why. It is necessary to show what is available and which data are missing. It should be explained why the data basis is complete for this study. Then the algorithms should be discussed by the same view: why you do what and why

this way can provide the expected results or answers to the hypothesis. The chapter Conclusions is a summary. A discussion or a chapter Discussions is missing. This is necessary to provide the relation of the study results to the overall knowledge given in the Introduction. What are new results? Why at the end of the Conclusions improvements of the understanding of pollution and meteorological conditions are followed? The paper addresses relevant scientific questions within the scope of ACP. The paper presents novel concepts, ideas, tools and data. The scientific methods and assumptions are valid and clearly outlined so that substantial conclusions are reached. The description of experiments and calculations allow their reproduction by fellow scientists. The results are sufficient to support the interpretations and conclusions. The quality of the figures is good. The figure captions should be improved so that these are understandable without the overall manuscript: measurement methods and calculation methods should be given. The related work is well cited so that the authors give proper credit to related work and own new contribution. The title reflects the whole content of the paper. The abstract must be improved: It is too long and includes descriptions of processes which are from the Introduction. What are the objectives, measurement methods, data, analyses methods, new results and conclusions? The overall presentation is well structured and clear. The language is fluent and precise but must be improved in very much details. It is necessary that a native speaker is improving the manuscript. The mathematical formulae, symbols, abbreviations, and units are generally correctly defined and used. No parts of the paper (text, formulae, figures, tables) should be reduced, combined, or eliminated. The number and quality of references is appropriate. Specific Comments Sometimes air pollution and sometime aerosols are used for the same matter. Why? How the map of PM2.5 in Figure 4 is determined from the available network of monitoring stations? The same question is for Figure 3, 5, 7, 9. Why the vertical scale in Figure 12 and 13 from m in the figures before into hPa? Technical corrections The references are incomplete in lines 474, 497, 540, 546. Sometimes p. and sometimes pp. is used?

---

## Referee Comment (RC2) · Anonymous Referee #2 · 26 Dec 2020

To date, the fine-resolution structures of thermodynamic and dynamic properties of the PBL remains poorly quantified, which in turn impairs our understanding of the formation mechanism of frequently occurred air pollution episodes in developing countries like China and India. The manuscript by Jiang et al. revealed detailed dynamical-thermal structures of PBL in Beijing-Tianjin-Hebei region of China based on the atmospheric profiles from a variety of ground-based remote sensing instruments, meteorological measurements from AWS, and reanalysis, combined with objectively classified synoptic patterns. A novel mechanism considering the synergistic effect of synoptic pattern

and PBL is proposed, which makes sense to me. The analysis methods are scientific sound, and the manuscript is well organized. Nevertheless, some of the results interpretations are not crystal clear, several conclusions drawn here can not be adequately supported by the results. Therefore, this work has to be returned to the authors for revision before it can be accepted for publication in ACP. My comments are listed as below.

Major comments: 1. Section 2.2. There are several ground-based remote sensing instruments used here. The retrieval of atmospheric thermal and dynamic variables will inevitably incur some kinds of uncertainties or even errors from these instruments. Nevertheless, I can not find any discussion on the uncertainties. 2. Section 2.3: How many PM2.5 data were used for the classification of synoptic pattern? And what is the spatial distribution of 68 PM2.5 monitoring stations? both of which should be clarified in this part. 3. Section 2.4: the authors are suggested to make it clear what kind of measurements has the Richardson number method been applied to? 4. Figure 12 & L377-381: Divergence profile and vertical velocity show large difference. For instance, the lower troposphere dominated by convergence at all times of day, while only during daytime the vertical velocity is positive (does the negative value denotes updraft? Please clarify it in Figure 12 caption). The authors may explain the discrepancy between the profiles during different times of day for vertical velocity and divergence. Besides, "900 hPa" is not exact, either. 5. Figure 13 & L402-403: the regional breezes within the PBL is generally observed in daytime instead of nighttime, so I am curious of how the mechanism (cold air mass induced by breeze overlaid by warm advected air) work out in BTH during nighttime? Regarding the schematic in Fig. 13, PBLHtop and PBLHlow are not logically right, and can be revised to PBLtop and PBLlow. Besides, this schematic should focus on the BTH region where the findings apply only from this work.

Minor comments: 1. L37-38: "are dominated" -> "dominate" 2. L49-52: One important factor, PBL and its interaction with aerosol, is missing for accounting for the

frequently occurred atmospheric pollution episodes. This is relevant to the topic of this study. The author may consider citing the review paper by Li et al. 2017 (doi: 10.1093/nsr/nwx117) and related observational studies such as Ding et al., 2016 (doi:10.1002/2016GL067745); Lou et al., 2019 (doi:10.1029/2019EA000620), Petaja et al. 2016 (doi: 10.1038/srep18998), among others. 3. L63: "on"-> "to" 4. L85-86: "which acted as a lid and capped the pollution in the boundary layer" needs reference support, the authors can refer to Xu et al. 2019 (doi:10.1016/j.scitotenv.2018.08.088) 5. L115: "observation data provided"->"weather station operated" 6. L256: it is inappropriate to say "meridional winds turned to easterly". First of all, the authors are suggested to make it clear the meridional wind is northerly or southerly. Secondly, the horizontal location in Figure 5 and vertical location in Figure 6 are suggested to be clarified. Last, the hours or time should be specified as well. Otherwise, the authors can not well follow what the authors are talking about. 7. L256: "advective temperature inversion occurred from 600 to 900 m (Fig. 6d)": I can not see any temperature inversion layer located within 600 – 900 m a.g.l.. If my understanding is right, the temperature inversion only occurred at 08-09 LT on October 22 and early morning (00-11) of October 23, but not at altitudes ranging from 600-900 m. 8. L256-257: Again. I am confused with "accompanied by stable stratification (Fig. 6e)" . Please clarify when and where stratification occurred. Can you directly identify a stratification layer from Fig. 6e. Probably the authors need to expand the description and give a more clear interpretation with Fig. 6e. 9. L441: grammar errors in "On the other hand, regulate"

---

## Author Comment (AC1) · 27 Jan 2021

We thank the Reviewer #1 for the helpful suggestions on improving the manuscript. The responses are shown below:

1. The introduction is missing the definition of objectives of the paper and maybe some hypothesis of answers for questions. Response 1: Thanks for your suggestion. We have reorganized the Introduction and added the objectives in the last paragraph of Introduction as follows: To sum up, because of the unique topography and geographic

[Figure]

location of Beijing, large-scale circulation and regional-scale thermodynamic circulation both have appreciable impacts on PBL and air pollution. What are the characteristics of PBL structure and the temporal and spatial distribution of pollution under different circulation types, and how do the multiscale circulations jointly force the PBL structure to change when they coexist are still unrevealed. Therefore, one objective of this study is to investigate the PBL dynamic-thermal structure and the distribution of pollution area under the most frequent circulation types in Beijing. The other primary objective is to further explore the synergetic effects of multiscale circulations on PBL and pollution in detail.

2. The description of methods is missing an overall statement which data are required and why. It is necessary to show what is available and which data are missing. It should be explained why the data basis is complete for this study. Then the algorithms should be discussed by the same view: why you do what and why this way can provide the expected results or answers to the hypothesis. Response 2: Thank you for the suggestions, we have revised the manuscript. The available data, why the data basis is complete for the study, and the weather typing approach are stated as follows: Since the weather typing approach is able to classify the synoptic circulations into different types and the high vertical resolution remote sensing observations can measure the fine dynamic-thermal structures of PBL, the objectives can be achieved by employing weather typing approach and remote sensing measurements as a necessary first step. We systematically probed the PBL structure with multiple remote sensing devices including ceilometer, Doppler Lidar and microwave radiometer (MWR) from 2018 to 2019 in Beijing. The measuring location is 39.6°N and 116.2°E, in the courtyard of the Institute of Atmospheric Physics, Chinese Academy of Sciences (Fig. 1b). The origional and timely remote sensing data, with high temporal and spatial resolution, are fully capable to show the fine PBL dynamic-thermal structure and lay a foundation for revealing the innovative findings. In addition, winds from hundreds of automatic weather stations can characterize the fine horizontal flow field completely. Thus, the synergetic effects of multiscale circulations on PBL dynamic-thermal structures and air pollution

[revised manuscript text omitted]

4. The paper addresses relevant scientific questions within the scope of ACP. The paper presents novel concepts, ideas, tools and data. The scientific methods and assumptions are valid and clearly outlined so that substantial conclusions are reached. The description of experiments and calculations allow their reproduction by fellow scientists. The results are sufficient to support the interpretations and conclusions. The quality of the figures is good. The figure captions should be improved so that these are understandable without the overall manuscript: measurement methods and calculation methods should be given. Response 4: Thank you for the suggestions. We have revised the figure captions all over again, and added the description of measurement methods to express more clearly. More detailed calculation methods and steps are given in the Section 2. Data and Method.

5. The related work is well cited so that the authors give proper credit to related work and own new contribution. The title reflects the whole content of the paper. The abstract must be improved: It is too long and includes descriptions of processes which are from the Introduction. What are the objectives, measurement methods, data, analyses methods, new results and conclusions? Response 5: Thank you for the suggestions. We reorganized Abstract to make it more concise. The objectives, measurements

and analysis method have been supplemented. The new results about the synergetic effects of multiscale circulations on PBL and pollution, and the characteristics of PBL dynamic-thermal structure and the distribution of pollution area under typical circulations of Beijing are involved as follows: Both synoptic circulations and regional circulations play important roles in regulating planetary boundary layer (PBL) dynamic-thermal structure and air quality in complex ways. However the synergistic impacts of coexisting multiscale circulations aren't well understood. Such coupling mechanisms and the PBL structure dominated by typical circulations types are investigated in the Beijing megacity based on Lamb-Jenkinson weather typing approach and fine remote sensing measurements. The direct regulatory effect of synoptic circulations played a leading role in the daytime by transporting and accumulating pollutants in front of mountains. While in the nighttime synoptic-scale and regional-scale circulations synergistically worsen the pollution. At night, the horizontal coupling mechanism of multiscale circulations produces a pollution convergent zone of different direction winds, while the vertical coupling mechanism regulates the mixing and diffusion of pollutants by changing the PBL dynamic-thermal structure. The warm advection transported by upper environmental winds overlies the cold advection transported by lower regional breezes, generating strong wind direction shear and advective inversion. The capping inversion and the convergent sinking motion within the PBL suppress massive pollutants below the zero-speed zone. The PBL dynamic-thermal structures under different typical synoptic circulations varies a lot. Under cyclonic circulation, the multilayer PBL is characterized by high vertical shear (600 m), temperature inversion (900 m) and an inhomogeneous stratification with sharp jump of Richardson number. The severe pollution zone is located to the south of Beijing. Under southwesterly circulation, the mono-layer PBL is characterized by low vertical shear (400 m) and inversion (200 m). Under westerly circulation, the PBL has a hybrid structure and the inversion is generated by the vertical shear of zonal winds. The pollution convergent zones under southwesterly and westerly circulations are located more northerly. There is no distinct PBL structure under anticyclone circulation.

The overall presentation is well structured and clear. The language is fluent and precise but must be improved in very much details. It is necessary that a native speaker is improving the manuscript. The mathematical formulae, symbols, abbreviations, and units are generally correctly defined and used. No parts of the paper (text, formulae, figures, tables) should be reduced, combined, or eliminated. The number and quality of references is appropriate.

Specific Comments

6. Sometimes air pollution and sometime aerosols are used for the same matter. Why? Response 6: Thanks for your comments very much. Since the Doppler lidar and ceilometer used in this study retrieved the 3D wind fields and attenuated backscatter coefficients by detecting the signals scattered by aerosols (ice, ash, dust, smoke), it is not appropriate to use pollution instead of aerosols sometimes. However the air pollution we studied in this paper refers to PM2.5 pollution dominated by aerosol particulates, therefore we use air pollution sometimes and use aerosols sometimes.

7. How the map of PM2.5 in Figure 4 is determined from the available network of monitoring stations? The same question is for Figure 3, 5, 7, 9. Response 7: Thanks for your suggestions. The details of pollution concentration in Fig. 4, 5, 7 and 9 have been added in Section 2.3 and the monitoring sites have been demonstrated in Fig. 4a: The hourly PM2.5 concentrations in the Beijing-Tianjin-Hebei monitoring sites are acquired from the National Urban Air Quality Real-time Publishing Platform (http://106.37.208.233:20035/) issued by the Ministry of Ecology and Environment. There are 35 air quality monitoring stations in Beijing (Fig. 4a) and 68 monitoring sites in Tianjin and Hebei provinces (Fig. 5, 7, 9). The PM2.5 concentration in Beijing are shown in shaded by interpolating data of 35 sites, while the PM2.5 concentration in other areas are shown in scatter with color as the spatial resolution is relative low. The PM2.5 data of Olympic Center station which is the closest monitoring site to the remote sensing measurements location is used in the circulation classification.

Fig. 4 The averaged PM2.5 concentration (shaded, units: 10-1 $\mu$g m-3) in Beijing for types C (a), SW (b), W (c) and A (d) from 2018 to 2019. The green dots in Fig. 4a indicate the locations of air quality monitoring sites in Beijing.

8. Why the vertical scale in Figure 12 and 13 from m in the figures before into hPa? Response 8: Thanks for your comments. Figure 6, 8 and 10 used to show the PBL dynamic-thermal structure measured by ceilometer, Doppler Lidar and Microwave Radiometer, and the remote sensing data was on the height levels, thus we adopted meter as the units inside the PBL. Figure 12 and 13 aimed to explain the vertical motion in the mid-low troposphere and its impact on PBL structure, and the reanalysis data was on the pressure levels, thus we adopted hPa as the units of the vertical coordinates. To demonstrate the corresponding height levels to the pressure levels, we added the height coordinate in Figure 12 and 13 as shown below:

Fig. 12 The averaged vertical velocity (units: Pa s-1, negative (positive) value denotes updraft (downward) movement) (a, c) and divergence (units: 10-5 s-1) (b, d) of pollution types (a, b) and the clean type (c, d) in the North China Plain

Technical corrections

9. The references are incomplete in lines 474, 497, 540, 546. Sometimes p. and sometimes pp. is used? Response 9: Thanks for your suggestions very much. The format of references has been modified all over again.

We thank the Reviewer #2 for the helpful suggestions on improving the manuscript. The responses are shown below:

1. Section 2.2. There are several ground-based remote sensing instruments used here. The retrieval of atmospheric thermal and dynamic variables will inevitably incur some kinds of uncertainties or even errors from these instruments. Nevertheless, I cannot find any discussion on the uncertainties. Response 1: Thank you for your suggestion. We have supplement the discussion on the uncertainties of remote sensing

measurements as follows: A full overlap is achieved by using the same telescope for transmitting and receiving so that the backscatter can be used from the first range gate (Münkel et al, 2007). This gives a clear advantage over other commonly used Automatic Lidars and Ceilometers that usually show great uncertainty in the range below 200–500 m (Kotthaus et al., 2018). Three possible PBL heights, with a temporal resolution of 10 minutes, can be output simultaneously to characterize the multiple aerosol layers structure according to the first three largest negative gradients of backscatter. The typical uncertainty of CL31 on attenuated backscatter coefficient is $\pm 20$ % and is $\pm 200$ m on mixing height determination compared with radiosonde and other active remote sensors (Tsaknakis et al., 2011). The velocity uncertainty along each LOS is associated with carrier-to-noise ratio (CNR) for each measurement volume following the methodology from O'Connor et al. (2010). Typically, a threshold of $-22$ or $-23$ dB is used as a limit for the accepted uncertainty in the lidar measurements (Gryning et al., 2016), which corresponds to an uncertainty of about 0.15 m s$-1$ (Aitken et al., 2012; Suomi et al., 2017). The MWR used in this study has been tested by comparing with radiosonde observations (Zhao et al., 2019). The systematic errors increase with altitude, and the MWR-retrieved temperature and relative humidity are of quite high reliability inside the PBL. The temperature biases and RMSEs are -2-0 °C and 1-2 °C under 2 km, and the minimum of biases and RMSEs are between 1 km and 2 km, less than 0.5 °C and 1.3 °C respectively. Since the relative humidity derived from the temperature and water vapor density, both the errors can cause the uncertainties. The bias and RMSE of relative humidity is about -5% and 15% under 2 km.

2. Section 2.3: How many PM2.5 data were used for the classification of synoptic pattern? And what is the spatial distribution of 68 PM2.5 monitoring stations? both of which should be clarified in this part. Response 2: Thank you for pointing this out. The details of PM2.5 data used for the classification of synoptic pattern have been added in Section 2.3 and the monitoring sites have been demonstrated in Fig. 4a: The hourly PM2.5 concentrations in the Beijing-Tianjin-Hebei monitoring sites are acquired from the National Urban Air Quality Real-time Publishing Platform

(http://106.37.208.233:20035/) issued by the Ministry of Ecology and Environment. There are 35 air quality monitoring stations in Beijing (Fig. 4a) and 68 monitoring sites in Tianjin and Hebei provinces (Fig. 5, 7, 9). The PM2.5 concentration in Beijing are shown in shaded by interpolating data of 35 sites, while the PM2.5 concentration in other areas are shown in scatter with color as the spatial resolution is relative low. The PM2.5 data of Olympic Center station, which is the closest monitoring site to the location of remote sensing measurements (less than 1 km), is used in the circulation classification.

Fig. 4 The averaged PM2.5 concentration (shaded, units: 10-1 $\mu$g m-3) in Beijing for types C (a), SW (b), W (c) and A (d) from 2018 to 2019. The green dots in Fig. 4a indicate the locations of air quality monitoring sites in Beijing.

3. Section 2.4: the authors are suggested to make it clear what kind of measurements has the Richardson number method been applied to? Response 3: Sorry for confusing you, we have added supplement description in Section 2.4 Method and Section 3.2 the discussion about the PBL structure as follows: The gradient Richardson number (Ri) is the ratio of the buoyancy term to the shear term in the turbulent kinetic equation. A negative Ri is an indication of buoyancy-generated turbulence, while positive Ri less than 0.25 indicates shear turbulence and dynamic instability. When Ri is larger than 0.25 and less than 1.0 the flows become neutral, or exhibit hysteresis and still maintain turbulent. Otherwise, Ri larger than 1.0 means turbulent flow will turn to be dynamically stable laminar (Stull, 1988). The distributional characteristics of Ri can reveal whether the PBL has a stratified structure or not (Banakh et al., 2020). Thus, we adopt the critical values of 0.25 and 1.0 as a criterion to determine the PBL structure. Ri can be calculated by Equation 1, where g is the acceleration of gravity and is the height interval between adjacent layers. is the mean virtual potential temperature, and is the mean zonal and meridional wind speeds within the height interval respectively. (1) The Richardson number Ri away from the temperature inversion structure was less than 0.25 (turbulent region) during the night, while it increased considerably

from the periphery of inversion and was larger than 1.0 (stable region) promptly. The sharp jump of Ri from the turbulent region to the stable region of inversion indicated a vertical stratified structure inside the PBL. The result suggested that the nocturnal PBL has an inhomogeneous stratification structure characterized by strong variations of Ri accompanied by inversion structure (Fig. 6e).

4. Figure 12 & L377-381: Divergence profile and vertical velocity show large difference. For instance, the lower troposphere dominated by convergence at all times of day, while only during daytime the vertical velocity is positive (does the negative value denotes updraft? Please clarify it in Figure 12 caption). The authors may explain the discrepancy between the profiles during different times of day for vertical velocity and divergence. Besides, "900 hPa" is not exact, either. Response 4: Thanks for your suggestion. We have supplemented the instruction in the caption of Fig. 12 and the explanation for the discrepancy between the profiles during different times of day in Section 3.3 as follows: In addition to horizontal circulations, the vertical motion of basic airflows is also a crucial dynamic factor in forming stable structure during pollution episodes. The pollution types shared similar vertical motion characteristics as shown in Fig. 12. The basic flows at the bottom of the troposphere is convergence and the flows above it is divergence at all times of a day (Fig. 12b). In the daytime, the environmental southerly winds ware obstructed on three sides by mountains. Airflows slowed down or stagnated in the plain areas, forming the topographic convergence. While at night the convergence was caused by the joint of environmental winds and regional breezes, and the height of convergence zone reduced simultaneously with the nocturnal PBL because the regional circulations developed below the shallower nocturnal PBL. Unlike the divergence field, the vertical velocity in the daytime differed from that in the nighttime because of the diurnal variations of PBL structure (Fig. 12a). In the daytime, the thermodynamic convection and the wind speed were enhanced expressively (Fig. 8a, 10a), thus the intensified turbulence will help the flows to move upward and cause the pollutants close to the ground to mix vertically within the PBL to some extent. However, the sinking and divergent flows superposed above the PBL, preventing

the pollutants from moving upward continuously and making it difficult for the aerosol particulates to diffuse beyond. As a consequence, the pollutants accumulate slowly in the daytime because of the common influences of horizontal topographic blocking and vertical upward mixing with the increasing PBL. However in nighttime, as the thermodynamic convection weakened and the inversion structure formed, it turned to be sinking movement at the bottom of the troposphere when the cold northerly regional breezes prevailed. Wu et al. (2017) found that the descending motion of synoptic circulations contributed to a reduction in the PBLH by compressing the air mass. Therefore, massive pollutants were capped near the surface and accumulated rapidly at night under the convergent sinking motion accompanied by temperature inversion structure.

Fig. 12 The averaged vertical velocity (units: Pa s-1, negative (positive) value denotes updraft (downward) movement) (a, c) and divergence (units: 10-5 s-1) (b, d) of pollution types (a, b) and the clean type (c, d) in the North China Plain

5. Figure 13 & L402-403: the regional breezes within the PBL is generally observed in daytime instead of nighttime, so I am curious of how the mechanism (cold air mass induced by breeze overlaid by warm advected air) work out in BTH during nighttime? Regarding the schematic in Fig. 13, PBLHtop and PBLHlow are not logically right, and can be revised to PBLtop and PBLlow. Besides, this schematic should focus on the BTH region where the findings apply only from this work. Response 5: Thanks for your suggestions. Yes, you are right. The regional breezes within the PBL are generally observed in the nighttime, thus the coupling mechanism of synoptic circulation and regional breezes works effectively during the night. In the daytime, large-scale southerly winds dominated and were favorable for transporting pollutants to north in the early stage of pollution. However during the period of pollution, the relative stronger horizontal winds and the increasing PBL make the pollution accumulate slowly or even remove the pollutants (Fig. 9d, 9e). We have reorganized the expression in Section 4. Conclusions and Discussion as follows: This paper explores the direct regulatory effect and indirect coupling effect of synoptic circulations by choosing the most frequent pollution

types and clean type classified by LWT approach. The PBL dynamic-thermal structure and the severe pollution area under typical circulations types are further investigated. Results suggest that different pollution patterns have similar influential mechanisms on PBL structure and air pollution. The direct regulatory effect of synoptic circulations plays a leading role in the daytime, large-scale southerly winds dominate and are favorable for the pollution transport to NCP region and the accumulation in front of mountains in the early stage of pollution. However during the period of pollution, the relative stronger southerly winds and the increasing PBL height are adverse to the accumulation of pollutants, or even make pollutants ventilated horizontally and diluted vertically. While the indirect effect played a leading role in the nighttime by coupling mechanisms. The coexisting multiscale circulations at night, on the one hand, affect the pollution via the horizontal coupling effect, which produces a pollution convergent zone of different direction winds. The relative strength of winds makes the polluted area move around horizontally between 39°N and 41°N. On the other hand, the multiscale circulations regulate the mixing and diffusion of pollutants by the vertical coupling effect, which changes the PBL dynamic and thermal structure. Vertical shear between the ambient winds and regional-scale breezes leads to advective inversion structure with strong variations of Ri. The nocturnal shallower PBL is consistent with the zero velocity zone, where massive pollutants were suppressed below, and the relative strength of winds determines the PBL height. The inaccurate expression and the horizontal part of schematic have been modified as follows:

Fig. 13 The schematic of vertical coupling mechanism of multiscale circulations for typical pollution types; The horizontal part is the background circulations of MSLP. The vertical part is the PBL dynamic-thermal structure over the NCP region

Minor comments

1. L37-38: "are dominated" -> "dominate" Response 1: Sorry for the mistake, we have corrected this.

2. L49-52: One important factor, PBL and its interaction with aerosol, is missing for accounting for the frequently occurred atmospheric pollution episodes. This is relevant to the topic of this study. The author may consider citing the review paper by Li et al. 2017 (doi:10.1093/nsr/nwx117) and related observational studies such as Ding et al., 2016 (doi:10.1002/2016GL067745); Lou et al., 2019 (doi:10.1029/2019EA000620), Petaja et al. 2016 (doi: 10.1038/srep18998), among others. Response 2: Thanks for the suggestion, the manuscript has been revised as below: In turn, the particulate matter can also affect the PBL structure by scattering and absorbing of solar radiation, and lead to severe pollution by positive feedback (Petaja et al, 2016; Li et al, 2017). Ding et al. (2016) suggested that black carbon enhanced haze pollution in megacities in China by heating upper PBL and cooling surface. Lou et al. (2019) investigated the relationships between PBL height and PM2.5 and indicated that the strongest anticorrelation occurred in the NCP region at 1400 Beijing time.

3. L63: "on"-> "to" Response 3: Sorry for the mistake, it has been modified.

4. L85-86:"which acted as a lid and capped the pollution in the boundary layer" needs reference support, the authors can refer to Xu et al. 2019 (doi:10.1016/j.scitotenv.2018.08.088) Response 4: Thanks for the suggestion, we have added the reference to support this viewpoint.

5. L115: "observation data provided"->"weather station operated" Response 5: Thank you for the correcting, the mistake has been corrected.

6. L256: it is inappropriate to say "meridional winds turned to easterly". First of all, the authors are suggested to make it clear the meridional wind is northerly or southerly. Secondly, the horizontal location in Figure 5 and vertical location in Figure 6 are suggested to be clarified. Last, the hours or time should be specified as well. Otherwise, the authors cannot well follow what the authors are talking about. Response 6: Thank you for pointing this out, we have corrected the mistake and specified the locations and time in the manuscript as follows: In the horizontal flow field, zonal winds from Tianjin

[Figure]

to the southeast of Beijing turned to be easterly winds and the northerly downslope winds in Beijing were strengthened later on the night of 22nd (Fig. 5b, 5c). Inside the PBL, easterly and northerly winds extended to 600 m above the ground from 20 pm on 22nd to 10 am on 23rd (Fig. 6b, 6c), thus the directional shear of meridional and zonal winds increased considerably.

7. L256: "advective temperature inversion occurred from 600 to 900 m (Fig. 6d)": I cannot see any temperature inversion layer located within 600 – 900 m. If my understanding is right, the temperature inversion only occurred at 08-09 LT on October 22 and early morning (00-11) of October 23, but not at altitudes ranging from 600-900 m. Response 7: Yes, your understanding is right, we have revised the description to clarify the expression more clearly as follows: Consequently, a conspicuous advective temperature inversion occurred near the shallower nocturnal PBL at 08-09 am on 22nd and 00-11 am on 23, ranging from 600 m to 900 m above the ground (Fig. 6d).

8. L256-257: Again. I am confused with "accompanied by stable stratification (Fig. 6e)". Please clarify when and where stratification occurred. Can you directly identify a stratification layer from Fig. 6e. Probably the authors need to expand the description and give a more clear interpretation with Fig. 6e. Response 8: Sorry for confusing you, we have revised the manuscript as answered in Response 3.

9. L441: grammar errors in "On the other hand, regulate" Response 9: Sorry for the mistakes, we have corrected this.
* * *
[Figure]

**Fig. 1.** Fig. 4 The PM2.5 concentration (shaded, units: 10-1 $\mu$g m-3) for types C (a), SW (b), W (c) and A (d). The green dots in Fig. 4a indicate the locations of air quality monitoring sites in Beijing

**Fig. 2.** Fig. 12 The vertical velocity (units: Pa s-1, negative (positive) denotes updraft (down-ward) movement) (a, c) and divergence (units: 10-5 s-1) (b, d) of pollution types (a, b) and clean type (c, d)

**Stratified structure.**

**Fig. 3.** Fig. 6 The gradient of temperature T'(z) (shaded, units: K km-1) measured by MWR (d), and Richardson number (shaded) (e) for type C

[Figure]

**Fig. 4.** Fig. 13 The schematic of vertical coupling mechanism for pollution types; The horizontal part is the background circulations. The vertical part is the PBL dynamic-thermal structure over NCP region

---

## Author Response (AR4)

**General comments**

The impression of the whole paper is a description of mechanisms of coupling atmospheric dynamics and air pollution levels in the North China Plain (NCP) and especially in the Beijing area.

We thank the Reviewer #1 for the helpful suggestions on improving the manuscript. The responses are shown below:

1. The introduction is missing the definition of objectives of the paper and maybe some hypothesis of answers for questions.

   Response 1: Thanks for your suggestion. We have reorganized the Introduction and added the objectives in the last paragraph of Introduction as follows:

   *To sum up, because of the unique topography and geographic location of Beijing, large-scale circulation and regional-scale thermodynamic circulation both have appreciable impacts on PBL and air pollution. What are the characteristics of PBL structure and the temporal and spatial distribution of pollution under different circulation types, and how do the multiscale circulations jointly force the PBL structure to change when they coexist are still unrevealed. Therefore, one objective of this study is to investigate the PBL dynamic-thermal structure and the distribution of pollution area under the most frequent circulation types in Beijing. The other primary objective is to further explore the synergetic effects of multiscale circulations on PBL and pollution in detail.*

2. The description of methods is missing an overall statement which data are required and why. It is necessary to show what is available and which data are missing. It should be explained why the data basis is complete for this study. Then the algorithms should be discussed by the same view: why you do what and why this way can provide the expected results or answers to the hypothesis.

   Response 2: Thank you for the suggestions, we have revised the manuscript carefully referred to your suggestions in Section 2. Data and Method as follows:

   *A PBL field observation experiment was performed from January 2018 to December 2019 basing on multiple remote sensing devices, including Doppler Wind Profile Lidar, microwave radiometer (MWR) and ceilometer in the courtyard of the Institute of Atmospheric Physics (39.6°N and 116.2°E), Chinese Academy of Sciences, Beijing (Fig. 1b). We systematically probed the PBL dynamic structure, thermodynamic structure and the vertical distribution of aerosols using the Lidar three-dimensional winds, the MWR temperature and humidity profiles and the ceilometer backscattering coefficient respectively. The original remote sensing data, with high temporal and spatial resolution, are fully capable to show the fine PBL dynamic-thermal structure. The reanalysis data of mean sea level pressure (MSLP) and winds are used to depict the synoptic circulations, and winds from hundreds of automatic weather stations to characterize the fine regional circulations. Thus, the synergistic*

*impacts of coexisting synoptic-scale and regional-scale circulations on the PBL dynamic-thermal structure and air pollution in Beijing megacity can be well understood using the remote sensing and meteorological data in combination with the Lamb-Jenkinson weather typing approach. The typical cases lasting two days in the same weather type (C, SW, W and A) are on October 22 to 24, July 26 to 28, May 15 to 17 in 2019 and December 28 to 30 in 2018 respectively. Due to the algorithm limitations on the observation conditions, the data of backscattering coefficient and temperature profiles are missing about 5 hours on July 27, 2019.*

The reasons for missing data are clarified in Section 2.1 Remote sensing data:

*The BL-VIEW algorithm excluded profiles with fog, precipitation or low clouds, therefore resulting in the missing value of attenuated backscatter coefficient on July 27, 2019 used in southwesterly circulation.*

*Furthermore, the residual liquid droplets on the water film led to high brightness temperature measured by the MWR, resulting in the abnormal high values of the temperature and humidity data. Therefore, data on July 27, 2019 were eliminated and substituted with missing values.*

3. The chapter Conclusions is a summary. A discussion or a chapter Discussions is missing. This is necessary to provide the relation of the study results to the overall knowledge given in the Introduction. What are new results? Why at the end of the Conclusions improvements of the understanding of pollution and meteorological conditions are followed?

Response 3: Thank you for the comments very much. The Section 4. Conclusions has been reorganized. Two aspects of new findings and the answers to the objectives, including the PBL dynamic-thermal structure and air pollution under typical circulations types, and the direct regulatory effect and indirect coupling effect of synoptic circulations on them are discussed in Section 4. Conclusions and Discussion.

[revised manuscript text omitted]

4. The paper addresses relevant scientific questions within the scope of ACP. The paper presents novel concepts, ideas, tools and data. The scientific methods and assumptions

are valid and clearly outlined so that substantial conclusions are reached. The description of experiments and calculations allow their reproduction by fellow scientists. The results are sufficient to support the interpretations and conclusions. The quality of the figures is good.

The figure captions should be improved so that these are understandable without the overall manuscript: measurement methods and calculation methods should be given.

Response 4: Thank you for the suggestions. We have revised the figure captions all over again, and added the description of measurement methods to express more clearly. More detailed calculation methods and steps are given in the Section 2. Data and Method.

5. The related work is well cited so that the authors give proper credit to related work and own new contribution. The title reflects the whole content of the paper. The abstract must be improved: It is too long and includes descriptions of processes which are from the Introduction. What are the objectives, measurement methods, data, analyses methods, new results and conclusions?

Response 5: Thank you for the suggestions. We reorganized Abstract to make it more concise. The objectives, measurements and analysis method have been supplemented. The new results about the synergetic effects of multiscale circulations on PBL and pollution, and the characteristics of PBL dynamic-thermal structure and severe pollution area under typical circulations in Beijing are involved as follows:

*To investigate the impacts of multiscale circulations on the planetary boundary layer (PBL), we have carried out the PBL dynamic-thermal structure field experiment with a Doppler Wind Profile Lidar, a microwave radiometer and a ceilometer from January 2018 to December 2019 in Beijing. We found that the direct regulatory effect of synoptic circulation played a leading role in the daytime by transporting and accumulating pollutants in front of mountains. While the indirect effect of multiscale circulations played a leading role in the nighttime by coupling mechanisms. The horizontal coupling of different direction winds produced a severe pollution convergent zone. The vertical coupling of upper environmental winds and lower regional breezes regulated the mixing and diffusion of pollutants by generating dynamic wind shear and advective temperature inversion. We also found that the dominated synoptic circulations leaded to great differences in PBL dynamic-thermal structure and pollution. The cyclonic circulation resulted in a typical multilayer PBL characterized by high vertical shear (600 m), temperature inversion (900 m) and an inhomogeneous stratification. Meanwhile, strong regional breezes pushed the pollution convergent zone to the south of Beijing. The southwesterly circulation resulted in a mono-layer PBL characterized by low vertical shear (400 m) and inversion (200 m). The westerly circulation leaded to a hybrid-structure PBL, and the advective inversion generated by the vertical shear of zonal winds. Strong environmental winds of southwesterly and westerly circulations pushed the severe pollution zone to the front of*

*mountains. There was no distinct PBL structure under the anticyclone circulation. The study systematically revealed the appreciable effects of synoptic and regional circulations on PBL structure and air quality, which enriched the prediction theory of atmospheric pollution in the complex terrain.*

**The overall presentation is well structured and clear. The language is fluent and precise but must be improved in very much details. It is necessary that a native speaker is improving the manuscript. The mathematical formulae, symbols, abbreviations, and units are generally correctly defined and used. No parts of the paper (text, formulae, figures, tables) should be reduced, combined, or eliminated. The number and quality of references is appropriate.**

**Specific Comments**

6. Sometimes air pollution and sometime aerosols are used for the same matter. Why?

   Response 6: Thanks for your comments very much. Since the Doppler lidar and ceilometer used in this study retrieved the 3D wind fields and attenuated backscatter coefficients by detecting the signals scattered by aerosols (ice, ash, dust, smoke), it is not appropriate to use pollution instead of aerosols sometimes. However the air pollution we studied in this paper refers to PM2.5 pollution dominated by aerosol particulates, therefore we use air pollution sometimes and use aerosols sometimes.

7. How the map of PM2.5 in Figure 4 is determined from the available network of monitoring stations? The same question is for Figure 3, 5, 7, 9.

   Response 7: Thanks for your suggestions. The details of pollution concentration in Fig. 4, 5, 7 and 9 have been added in Section 2.3 and the monitoring sites have been demonstrated in Fig. 4a:

   *The hourly PM2.5 concentrations in the Beijing-Tianjin-Hebei monitoring sites are acquired from the National Urban Air Quality Real-time Publishing Platform (http://106.37.208.233:20035/) issued by the Ministry of Ecology and Environment. There are 35 air quality monitoring stations in Beijing (Fig. 4a) and 68 monitoring sites in Tianjin and Hebei provinces (Fig. 5, 7, 9). The PM2.5 concentration in Beijing are shown in shaded by interpolating data of 35 sites, while the PM2.5 concentration in other areas are shown in scatter with color as the spatial resolution is relative low. The PM2.5 data of Olympic Center station which is the closest monitoring site to the remote sensing measurements location is used in the circulation classification.*

[Figure]

Fig. 4 The averaged PM2.5 concentration (shaded, units: $10^{-1}$ µg m$^{-3}$) in Beijing for types C (a), SW (b), W (c) and A (d) from 2018 to 2019. The green dots in Fig. 4a indicate the locations of air quality monitoring sites in Beijing.

8. Why the vertical scale in Figure 12 and 13 from m in the figures before into hPa?
   Response 8: Thanks for your comments. Figure 6, 8 and 10 used to show the PBL dynamic-thermal structure measured by ceilometer, Doppler Lidar and Microwave Radiometer, and the remote sensing data was on the height levels, thus we adopted meter as the units inside the PBL. Figure 12 and 13 aimed to explain the vertical motion in the mid-low troposphere and its impact on PBL structure, and the reanalysis data was on the pressure levels, thus we adopted hPa as the units of the vertical coordinates. To demonstrate the corresponding height levels to the pressure levels, we added the height coordinate in Figure 12 and 13 as shown below:

[Figure]

Fig. 12 The averaged vertical velocity (units: Pa s⁻¹, negative (positive) value denotes updraft (downward) movement) (a, c) and divergence (units: 10⁻⁵ s⁻¹) (b, d) of pollution types (a, b) and the clean type (c, d) in the North China Plain

**Technical corrections**

9.  The references are incomplete in lines 474, 497, 540, 546. Sometimes p. and sometimes pp. is used?

    Response 9: Thanks for your suggestions very much. The format of references has been modified all over again.

**We thank the Reviewer #2 for the helpful suggestions on improving the manuscript. The responses are shown below:**

1. Section 2.2. There are several ground-based remote sensing instruments used here. The retrieval of atmospheric thermal and dynamic variables will inevitably incur some kinds of uncertainties or even errors from these instruments. Nevertheless, I cannot find any discussion on the uncertainties.

   Response 1: Thank you for your suggestion. We have supplement the discussion on the uncertainties of remote sensing measurements as follows:

   *A full overlap is achieved by using the same telescope for transmitting and receiving so that the backscatter can be used from the first range gate (Münkel et al, 2007). This gives a clear advantage over other commonly used Automatic Lidars and Ceilometers that usually show great uncertainty in the range below 200–500 m (Kotthaus et al., 2018). Three possible PBL heights, with a temporal resolution of 10 minutes, can be output simultaneously to characterize the multiple aerosol layers structure according to the first three largest negative gradients of backscatter. The typical uncertainty of CL31 on attenuated backscatter coefficient is ±20 % and is ±200 m on mixing height determination compared with radiosonde and other active remote sensors (Tsaknakis et al., 2011).*

   *The velocity uncertainty along each LOS is associated with carrier-to-noise ratio (CNR) for each measurement volume following the methodology from O'Connor et al. (2010). Typically, a threshold of −22 or −23 dB is used as a limit for the accepted uncertainty in the lidar measurements (Gryning et al., 2016), which corresponds to an uncertainty of about 0.15 m s⁻¹ (Aitken et al., 2012; Suomi et al., 2017).*

   *The MWR used in this study has been tested by comparing with radiosonde observations (Zhao et al., 2019). The systematic errors increase with altitude, and the MWR-retrieved temperature and relative humidity are of quite high reliability inside the PBL. The temperature biases and RMSEs are -2-0 °C and 1-2 °C under 2 km, and the minimum of biases and RMSEs are between 1 km and 2 km, less than 0.5 °C and 1.3 °C respectively. Since the relative humidity derived from the temperature and water vapor density, both the errors can cause the uncertainties. The bias and RMSE of relative humidity is about -5% and 15% under 2 km.*

2. Section 2.3: How many PM2.5 data were used for the classification of synoptic pattern? And what is the spatial distribution of 68 PM2.5 monitoring stations? both of which should be clarified in this part.

   Response 2: Thank you for pointing this out. The details of PM2.5 data used for the classification of synoptic pattern have been added in Section 2.3 and the monitoring sites have been demonstrated in Fig. 4a:

   *The hourly PM2.5 concentrations in the Beijing-Tianjin-Hebei monitoring sites are acquired from the National Urban Air Quality Real-time Publishing Platform (http://106.37.208.233:20035/) issued by the Ministry of Ecology and*

*Environment. There are 35 air quality monitoring stations in Beijing (Fig. 4a) and 68 monitoring sites in Tianjin and Hebei provinces (Fig. 5, 7, 9). The PM2.5 concentration in Beijing are shown in shaded by interpolating data of 35 sites, while the PM2.5 concentration in other areas are shown in scatter with color as the spatial resolution is relative low. The PM2.5 data of Olympic Center station, which is the closest monitoring site to the location of remote sensing measurements (less than 1 km), is used in the circulation classification.*

[Figure]

Fig. 4 The averaged PM2.5 concentration (shaded, units: $10^{-1}$ µg m$^{-3}$) in Beijing for types C (a), SW (b), W (c) and A (d) from 2018 to 2019. The green dots in Fig. 4a indicate the locations of air quality monitoring sites in Beijing.

3. Section 2.4: the authors are suggested to make it clear what kind of measurements has the Richardson number method been applied to?

    Response 3: Sorry for confusing you, we have added supplement description in Section 2.4 Method and Section 3.2 the discussion about the PBL structure as follows:

    *The gradient Richardson number (Ri) is the ratio of the buoyancy term to the shear term in the turbulent kinetic equation. A negative Ri is an indication of buoyancy-generated turbulence, while positive Ri less than 0.25 indicates shear turbulence and dynamic instability. When Ri is larger than 0.25 and less than 1.0 the flows become neutral, or exhibit hysteresis and still maintain turbulent. Otherwise, Ri larger than 1.0 means turbulent flow will turn to be dynamically stable laminar (Stull, 1988). The distributional characteristics of Ri can reveal whether the PBL has a stratified structure or not (Banakh et al., 2020). Thus, we adopt the critical values of 0.25 and 1.0 as a criterion to determine the PBL structure. Ri can be calculated by Equation 1, where g is the acceleration of gravity and $\Delta z$ is the height interval between adjacent layers.*

    *$\bar{\theta}$ is the mean virtual potential temperature, $\Delta\bar{u}$ and $\Delta\bar{v}$ is the mean zonal*

*and meridional wind speeds within the height interval respectively.*

$$R_i = \frac{\dfrac{g}{\bar{\theta}}\dfrac{\Delta\bar{\theta}}{\Delta z}}{(\dfrac{\Delta\bar{u}}{\Delta z})^2 + (\dfrac{\Delta\bar{v}}{\Delta z})^2} \quad (1)$$

*The Richardson number Ri away from the temperature inversion structure was less than 0.25 (turbulent region) during the night, while it increased considerably from the periphery of inversion and was larger than 1.0 (stable region) promptly. The sharp jump of Ri from the turbulent region to the stable region of inversion indicated a vertical stratified structure inside the PBL. The result suggested that the nocturnal PBL has an inhomogeneous stratification structure characterized by strong variations of Ri accompanied by inversion structure (Fig. 6e).*

[Figure]

4.  Figure 12 & L377-381: Divergence profile and vertical velocity show large difference. For instance, the lower troposphere dominated by convergence at all times of day, while only during daytime the vertical velocity is positive (does the negative value denotes updraft? Please clarify it in Figure 12 caption). The authors may explain the discrepancy between the profiles during different times of day for vertical velocity and divergence. Besides, "900 hPa" is not exact, either.
    Response 4: Thanks for your suggestion. We have supplemented the instruction in the caption of Fig. 12 and the explanation for the discrepancy between the profiles during different times of day in Section 3.3 as follows:

    *In addition to horizontal circulations, the vertical motion of basic airflows is also a crucial dynamic factor in forming stable structure during pollution episodes. The pollution types shared similar vertical motion characteristics as shown in Fig. 12. The basic flows at the bottom of the troposphere is convergence and the flows above it is divergence at all times of a day (Fig. 12b).*

*In the daytime, the environmental southerly winds ware obstructed on three sides by mountains. Airflows slowed down or stagnated in the plain areas, forming the topographic convergence. While at night the convergence was caused by the joint of environmental winds and regional breezes, and the height of convergence zone reduced simultaneously with the nocturnal PBL because the regional circulations developed below the shallower nocturnal PBL. Unlike the divergence field, the vertical velocity in the daytime differed from that in the nighttime because of the diurnal variations of PBL structure (Fig. 12a). In the daytime, the thermodynamic convection and the wind speed were enhanced expressively (Fig. 8a, 10a), thus the intensified turbulence will help the flows to move upward and cause the pollutants close to the ground to mix vertically within the PBL to some extent. However, the sinking and divergent flows superposed above the PBL, preventing the pollutants from moving upward continuously and making it difficult for the aerosol particulates to diffuse beyond. As a consequence, the pollutants accumulate slowly in the daytime because of the common influences of horizontal topographic blocking and vertical upward mixing with the increasing PBL. However in nighttime, as the thermodynamic convection weakened and the inversion structure formed, it turned to be sinking movement at the bottom of the troposphere when the cold northerly regional breezes prevailed. Wu et al. (2017) found that the descending motion of synoptic circulations contributed to a reduction in the PBLH by compressing the air mass. Therefore, massive pollutants were capped near the surface and accumulated rapidly at night under the convergent sinking motion accompanied by temperature inversion structure.*

[Figure]

Fig. 12 The averaged vertical velocity (units: Pa s$^{-1}$, negative (positive) value denotes updraft (downward)

movement) (a, c) and divergence (units: $10^{-5}$ $s^{-1}$) (b, d) of pollution types (a, b) and the clean type (c, d) in the North China Plain

5. Figure 13 & L402-403: the regional breezes within the PBL is generally observed in daytime instead of nighttime, so I am curious of how the mechanism (cold air mass induced by breeze overlaid by warm advected air) work out in BTH during nighttime? Regarding the schematic in Fig. 13, PBLHtop and PBLHlow are not logically right, and can be revised to PBLtop and PBLlow. Besides, this schematic should focus on the BTH region where the findings apply only from this work.

Response 5: Thanks for your suggestions. Yes, you are right. The regional breezes within the PBL are generally observed in the nighttime, thus the coupling mechanism of synoptic circulation and regional breezes works effectively during the night. In the daytime, large-scale southerly winds dominated and were favorable for transporting pollutants to north in the early stage of pollution. However during the period of pollution, the relative stronger horizontal winds and the increasing PBL make the pollution accumulate slowly or even remove the pollutants (Fig. 9d, 9e). We have reorganized the expression in Section 4. Conclusions and Discussion as follows:

*This paper explores the direct regulatory effect and indirect coupling effect of synoptic circulations by choosing the most frequent pollution types and clean type classified by LWT approach. The PBL dynamic-thermal structure and the severe pollution area under typical circulations types are further investigated. Results suggest that different pollution patterns have similar influential mechanisms on PBL structure and air pollution. The direct regulatory effect of synoptic circulations plays a leading role in the daytime, large-scale southerly winds dominate and are favorable for the pollution transport to NCP region and the accumulation in front of mountains in the early stage of pollution. However during the period of pollution, the relative stronger southerly winds and the increasing PBL height are adverse to the accumulation of pollutants, or even make pollutants ventilated horizontally and diluted vertically. While the indirect effect played a leading role in the nighttime by coupling mechanisms. The coexisting multiscale circulations at night, on the one hand, affect the pollution via the horizontal coupling effect, which produces a pollution convergent zone of different direction winds. The relative strength of winds makes the polluted area move around horizontally between 39°N and 41°N. On the other hand, the multiscale circulations regulate the mixing and diffusion of pollutants by the vertical coupling effect, which changes the PBL dynamic and thermal structure. Vertical shear between the ambient winds and regional-scale breezes leads to advective inversion structure with strong variations of Ri. The nocturnal shallower PBL is consistent with the zero velocity zone, where massive pollutants were suppressed below, and the relative strength of winds determines the PBL height.*

The inaccurate expression and the horizontal part of schematic have been modified as follows:

[Figure]

Fig. 13 The schematic of vertical coupling mechanism of multiscale circulations for typical pollution types; The horizontal part is the background circulations of MSLP. The vertical part is the PBL dynamic-thermal structure over the NCP region

**Minor comments**

1.  L37-38: "are dominated" -> "dominate"
    Response 1: Sorry for the mistake, we have corrected this.

2.  L49-52: One important factor, PBL and its interaction with aerosol, is missing for accounting for the frequently occurred atmospheric pollution episodes. This is relevant to the topic of this study. The author may consider citing the review paper by Li et al. 2017 (doi:10.1093/nsr/nwx117) and related observational studies such as Ding et al., 2016 (doi:10.1002/2016GL067745); Lou et al., 2019 (doi:10.1029/2019EA000620), Petaja et al. 2016 (doi: 10.1038/srep18998), among others.

    Response 2: Thanks for the suggestion, the manuscript has been revised as below:

    *In turn, the particulate matter can also affect the PBL structure by scattering and absorbing of solar radiation, and lead to severe pollution by positive feedback (Petaja et al, 2016; Li et al, 2017). Ding et al. (2016) suggested that black carbon enhanced haze pollution in megacities in China by heating upper PBL and cooling surface. Lou et al. (2019) investigated the relationships between PBL height and PM2.5 and indicated that the strongest anticorrelation occurred in the NCP region at 1400 Beijing time.*

3. L63: "on"-> "to"
   Response 3: Sorry for the mistake, it has been modified.

4. L85-86: "which acted as a lid and capped the pollution in the boundary layer" needs reference support, the authors can refer to Xu et al. 2019 (doi:10.1016/j.scitotenv.2018.08.088)
   Response 4: Thanks for the suggestion, we have added the reference to support this viewpoint.

5. L115: "observation data provided"->"weather station operated"
   Response 5: Thank you for the correcting, the mistake has been corrected.

6. L256: it is inappropriate to say "meridional winds turned to easterly". First of all, the authors are suggested to make it clear the meridional wind is northerly or southerly. Secondly, the horizontal location in Figure 5 and vertical location in Figure 6 are suggested to be clarified. Last, the hours or time should be specified as well. Otherwise, the authors cannot well follow what the authors are talking about.
   Response 6: Thank you for pointing this out, we have corrected the mistake and specified the locations and time in the manuscript as follows:
   ***In the horizontal flow field, zonal winds from Tianjin to the southeast of Beijing turned to be easterly winds and the northerly downslope winds in Beijing were strengthened later on the night of 22nd (Fig. 5b, 5c). Inside the PBL, easterly and northerly winds extended to 600 m above the ground from 20 pm on 22nd to 10 am on 23rd (Fig. 6b, 6c), thus the directional shear of meridional and zonal winds increased considerably.***

7. L256: "advective temperature inversion occurred from 600 to 900 m (Fig. 6d)": I cannot see any temperature inversion layer located within 600 – 900 m. If my understanding is right, the temperature inversion only occurred at 08-09 LT on October 22 and early morning (00-11) of October 23, but not at altitudes ranging from 600-900 m.
   Response 7: Yes, your understanding is right, we have revised the description to clarify the expression more clearly as follows:
   ***Consequently, a conspicuous advective temperature inversion occurred near the shallower nocturnal PBL at 08-09 am on 22nd and 00-11 am on 23, ranging from 600 m to 900 m above the ground (Fig. 6d).***

8. L256-257: Again. I am confused with "accompanied by stable stratification (Fig. 6e)". Please clarify when and where stratification occurred. Can you directly identify a stratification layer from Fig. 6e. Probably the authors need to expand the description and give a more clear interpretation with Fig. 6e.
    Response 8: Sorry for confusing you, we have revised the manuscript as

answered in Response 3.

9. L441: grammar errors in "On the other hand, regulate"
   Response 9: Sorry for the mistakes, we have corrected this.

**We thank the Editor for the helpful suggestions on improving the manuscript. The responses are shown below:**

1. Grammatical error in Lines 20-22 need be modified.
   Response 1: Sorry for the mistake. We modified the manuscript as follows:

   *We found that the direct regulatory effect of synoptic circulation worked through transporting and accumulating pollutants in front of mountains in the daytime. While the indirect effect of multiscale circulations worked through coupling mechanisms in the nighttime.*

2. "red line " in the description of Fig. 1 "The location of Beijing city in China" is inappropriate.
   Response 2: Thank you for the suggestion. We have revised the figure description as follows:

   *Fig. 1 The locations of 16 grid data of the 5°× 10° MSLP used for Lamb-Jenkinson weather type classification (black dots) (a). The terrain height of the North China Plain (shaded, units: m) (b). The location of Beijing city and the Beijing-Tianjin-Hebei region is marked by the red solid lines and red dashed lines respectively. The orange dot indicates the location of remote sensing devices. The arrows indicate the horizontal coupling mechanism of how multiscale circulations affect pollution by generating convergent zone.*

3. More studies on the association between circulation and PM2.5 pollution should be summarized in introduction or compared with results. (e.g. Leung et al., 2018, DOI: 10.5194/acp-2017-916; Liu et al., 2019, doi: 10.1007/s13351-019-9007-z).
   Response 3: Thanks for your suggestion. We have added researches on the association between circulation and PM2.5 pollution in Section Introduction as follows:

   *Leung et al. (2018) indicated that daily PM2.5 had strong positive correlation with temperature and relative humidity but negative correlation with sea-level pressure in northern China. The PM2.5-to-climate sensitivities results were applied to predict future PM2.5 due to climate change, and found a decrease of 0.5μg/m3 in annual mean PM2.5 in the Beijing-Tianjin-Hebei region due to more frequent cold frontal ventilation. Liu et al. (2019) found that the episodes of PM2.5 pollution over the Beijing-Tianjin-Hebei region in winter were related to weather patterns such as the rear of a high-pressure system approaching the sea, a high-pressure field, a saddle pressure field, and the leading edge of a cold front.*